# Characterization of Stratospheric Particle Size Distribution Uncertainties using SAGE II and SAGE III/ISS Extinction Spectra

Travis N. Knepp[1], Mahesh Kovilakam[1, 2], Larry Thomason[1,4], and Stephen J. Miller[3,4]

[1]NASA Langley Research Center, Hampton, Virginia 23681, USA
[2]ADNET Systems Inc, 6720B Rockledge Drive, Suite 504 Bethesda, MD 20817, USA
[3]Huntington Ingalls Industries Newport News, Virginia 23607, USA
[4]Retired

**Correspondence:** Travis N. Knepp (travis.n.knepp@nasa.gov)

**Abstract.** A new algorithm was developed to infer particle size distribution parameters from the Stratospheric Aerosol and Gas Experiment II (SAGE II) and SAGE III on the International Space Station (SAGE III/ISS) extinction spectra using a lookup table (LUT) approach. Here, the SAGE-based extinction ratios were matched to LUT values and, using these matches, weighted statistics were calculated to infer the median particle size distribution values and higher-moment parameters as well as quantify the uncertainty in these estimates. This was carried out by solving for both single and bimodal lognormal distributions. The work presented herein falls under 2 general headings: 1. a theoretical study was carried out to determine the accuracy of this methodology; 2. the solution algorithm was applied to the SAGE II and SAGE III/ISS records with a brief case study analysis of the 2022 Hunga Tonga eruption. This methodology was demonstrated to be ≈25% accurate for mode radius and has a minor dependence on particle composition. While bimodal solutions were obtained from this algorithm, we provide a conclusive demonstration of how and why these estimates are inherently unstable using SAGE III/ISS extinction spectra along. Finally, we demonstrated how the Hunga Tonga aerosol plume evolved in regard to both size and transport over 18 months after the 2022 Hunga Tonga eruption. The PSD estimates, higher-moment parameters, and uncertainties are new products within the SAGE III/ISS level 2 products, are currently available for download and will be merged into the main SAGE III/ISS release products in a subsequent L2 release.

## 1 Introduction

Stratospheric aerosols play a key role in determining the chemistry (Hofmann and Solomon, 1989; Fahey et al., 1993; Solomon et al., 1996) and radiative balance (Minnis et al., 1993; Ridley et al., 2014) of the atmosphere. Recent changes in the stratospheric aerosol loading (since 2000) have received significant attention in the scientific community (Hofmann et al., 2009; Vernier et al., 2011; Ridley et al., 2014; Santer et al., 2014; Vernier et al., 2015; Solomon et al., 2011; Bourassa et al., 2012). Many global climate models (GCMs) rely on observational data to represent stratospheric aerosols as these GCMs do not have an interactive stratospheric aerosol scheme to model aerosol properties (Kremser et al., 2016). An accurate representation of stratospheric aerosol properties including particle size distribution (PSD) is therefore important. While there are other global stratospheric aerosol measurements available since the early 1980s, the Stratospheric Aerosol and Gas Experiment (SAGE)

series of satellites (McCormick et al., 1979) have provided a long-term (1979 – 2005; 2017 – present), multi-wavelength, global record of stratospheric aerosols. To date, data collected by the SAGE family of instruments have been instrumental in improving our understanding of chemistry, radiative balance, and atmospheric dynamics (Lu et al., 2000; Fadnavis et al., 2013; Dube et al., 2020).

The second SAGE instrument (SAGE II) collected data between 1985 and 2005 and the current SAGE instrument on the International Space Station (SAGE III/ISS) began collecting data in June 2017. The measurement method of all SAGE instruments has been occultation: peering through the Earth's atmosphere to observe how the solar spectrum changes as a function of altitude. This allows retrieval of altitude-dependent number densities for gas-phase species (e.g. $O_3$, $NO_2$, $H_2O$) as well as a series of aerosol extinction coefficients (referred to, hereafter, as extinction), which have an expected precision on the order of $\approx$5-10% (NASA, 2018). The extinction spectra (i.e., how extinction changes as a function of wavelength) contain limited information regarding composition (Knepp et al., 2022) as well as the microphysical properties (e.g. particle size distributions) for the aerosol responsible for attenuating the Sun's light (von Savigny and Hoffmann, 2020; Wrana et al., 2021, 2023). These microphysical properties play a key role in regulating atmospheric chemistry (Hofmann and Solomon, 1989; Fahey et al., 1993; Solomon et al., 1996) and radiative balance (Minnis et al., 1993; Ridley et al., 2014) in GCMs. Indeed, extinction spectra have been used to infer aerosol surface area density (SAD) and effective radius ($r_e$) for the SAGE II instrument (Thomason et al., 1997, 2008; Damadeo et al., 2013), which was released as a standard product for that dataset. However, despite the high level of precision in the aerosol extinction product, the utility of the SAD product was predominantly limited to deriving SAD in the wake of volcanic events since, during background conditions, the uncertainty of the SAD product was potentially >200% (Thomason et al., 2008). While SAD and $r_e$ were released as part of the SAGE II product, PSD parameters (e.g., mode radius and distribution width) have never been part of the standard SAGE product. To date, external groups have been responsible for developing their own algorithms and methodologies for extracting this information from the extinction spectra (e.g. Wang et al., 1989a, b, 1996; Yue, 2000; Bingen et al., 2004a; Wrana et al., 2021, 2023). Such inferences are challenging and inherently unstable due to the ill-posedness of the problem (Fussen et al., 2001). One method to stabilize the solution space is to apply a smoothness condition that forces the PSD parameters to vary smoothly with altitude (Bingen et al., 2004a, b, 2006, 2017). While this methodology tends to produce reasonable values, it has not been evaluated to determine its accuracy or precision. Indeed, validating these products by comparison to in situ observations (e.g. from aircraft field campaigns or balloon flights) would be challenging. The challenge in performing "traditional" validation exercises (e.g., involving aircraft and balloon campaigns) that compare these derived products with in situ observations comes primarily from differing sampling volumes. As an occultation measurement, the SAGE instrument peers through hundreds of kilometers of atmosphere as the instrument scans across the solar disk. Therefore, the question becomes: how representative are the in situ measurements of the total sampling volume of the SAGE instrument. Generally speaking, such intercomparisons often gloss over this by assuming that the zonal variability of the observed species, over short time scales, is sufficiently low to allow intercomparison. While this has proven effective in validating ozone (Wang et al., 2020), similar treatment with aerosol has yet to consistently yield agreement better that 35% (Deshler et al., 2003) due, in part, to the differing sampling volumes and atmospheric heterogeneity.

An alternative to validating the inferred PSD parameters using empirical data is to gauge the potential accuracy of these methods by working in the other direction. By starting with Mie theory, where the microphysical properties are strictly defined, we can create lookup tables of extinction coefficients at SAGE wavelengths for varying combinations of PSD parameters. These lookup tables can then be used to identify all PSD parameters that yield matches to a series of SAGE-observed extinction coefficients (within the precision of the measurement). This will effectively provide the solution space of PSD and microphysical properties for SAGE extinction products. Ultimately, this results in a collection of PSD parameters for each SAGE profile, with a quantified error for the PSD estimate. This methodology is described below and is applied to the SAGE II and SAGE III/ISS record. The influence of common assumptions are discussed throughout the manuscript.

## 2   Instruments and data

The SAGE family instruments have been described previously (Mauldin et al., 1985; Cisewski et al., 2014). Briefly, SAGE instruments use the solar occultation method to measure the solar attenuation, as a function of wavelength, that occurs throughout the atmosphere. Standard products include the number density of gas-phase species (e.g., $O_3$, $NO_2$, and $H_2O$) as well as aerosol extinction coefficients (385, 450, 525, 1020 nm for SAGE II and 384, 448, 520, 601, 676, 755, 869, 1021, 1543 nm for SAGE III/ISS; referenced as $k_\lambda$). Herein, the SAGE II v7.0 and SAGE III/ISS v5.3 (June 2017 – November 2023, inclusive) products were used. All analysis was limited to altitudes between the tropopause and 30 km and the 601 nm and 676 nm channels were excluded from the analysis due to ozone interference within those channels (Wang et al., 2020).

### 2.1   Computing hardware and code

The analysis code was written in Python and relied heavily on the PyTorch (v1.12.1) library. Because of the nature of the lookup table (LUT) methodology and the data volume extensive parallelization was required to run the code within a reasonable time. This requirement was most pronounced during the bimodal analysis. Therefore, all of the solution routines as well as the statistical calculations were carried out on an NVIDIA A100 GPU with 80 GB of memory. While the 80 GB of memory within the A100 was required for the bimodal analysis, we note that all of the single-mode solution code could be run on a more modest GPU. Indeed, much of the development work was carried out on a Quadro RTX 3000.

## 3   Methodology

### 3.1   Lookup Table Construction

In this section we present a brief overview of the theoretical basis for constructing the LUTs as well as justification for the PSD boundaries used in generating these tables.

The processing time of this algorithm is directly related to the size of the LUTs. Further, the resolution and extent of the PSD parameters used to generate the LUTs directly controls the accuracy of the inferred PSD parameters. This creates a dilemma:

do we sacrifice accuracy for improved run time or improve the accuracy at the cost of run time? Therefore, after providing a general overview of the LUT creation we present brief justifications for the parameters and resolutions used herein.

Extinction coefficients at each SAGE wavelength ($k_\lambda$) were calculated using Mie theory under the assumption that all particles are spherical and the distribution is single-mode lognormal. This was done for sulfuric acid aerosol at different weight percents (65%, 70%, 75%, and 80% sulfuric acid by weight with water comprising the rest of the particle) using refractive indices reported by Palmer and Williams (1975), smoke composed of black carbon (BC) using refractive indices reported by Sumlin et al. (2018), and smoke composed of brown carbon (BrC) using refractive indices reported by Bergstrom et al. (2002). These refractive indices are presented in Table 1.

The PSD LUTs were generated by first calculating single particle extinction efficiencies ($\mathbf{Q_{ext}}(\boldsymbol{\lambda},\mathbf{r})$ where $\boldsymbol{\lambda}$ was an array of SAGE wavelengths and $\mathbf{r}$ was an array of particle radii (here, $\mathbf{r} = [10, 11, 12, \ldots, 9999, 10000]$ nm); for derivation of $\mathbf{Q_{ext}}(\lambda, \mathbf{r})$ see Kerker (1969), Hansen and Travis (1974), or Bohren and Huffman (1983). The extinction coefficients were then calculated by multiplying $\mathbf{Q_{ext}}$ with a series of single-mode lognormal distributions ($\mathbf{P}(r_m, \sigma)$; here, the number density (N) was set to 1 cm$^{-3}$) followed by integration (Eq. 1). The lognormal distribution is described in Eq. 2 where $\sigma$ is the geometric standard deviation (sometimes written $\sigma_g$ in the aerosol literature and hereafter referred to as distribution width) and $r_m$ is the mode radius (the median radius of a lognormal distribution is commonly referred to as mode radius in aerosol literature; we adopt this convention here). Lognormal distributions were calculated for mode radii ($r_m$) that extended from 10 nm through 1500 nm (1 nm resolution) and distribution widths that ranged from 1.01 through 2.0 with a resolution of 0.001. This resulted in three-dimensional lookup tables of extinction coefficients (i.e., $\mathbf{k}(\lambda, r_m, \sigma)$) as a function of incident light wavelength, mode radius, and distribution width.

$$\mathbf{k}(\lambda, r_m, \sigma) = \int\limits_{r_{min}}^{r_{max}} \pi \mathbf{r}^2 \mathbf{P}(r_m, \sigma) \mathbf{Q_{ext}}(\lambda, \mathbf{r}) \mathrm{d}r \tag{1}$$

$$\mathbf{P}(r_m, \sigma) = \frac{N}{\sqrt{2\pi} \ln(\sigma) \mathbf{r}} \exp\left[\frac{(\ln(\mathbf{r}) - \ln(r_m))^2}{-2\ln(\sigma)^2}\right] \tag{2}$$

Because the resolution and the range of particle radii and distribution widths used to create extinction tables plays a critical role in defining the accuracy of the LUTs, and the corresponding PSD parameters inferred from these LUTs, we provide a brief justification for the resolution and range of $r_m$ and $\sigma$ below.

### 3.1.1 Justification for extent of particle radii

The extent of particle radii used to create the extinction efficiency table as well as set the integration boundaries ($r_{min}$ and $r_{max}$ in Eq. (1)) was based on the influence large particles can have, despite their rarity, on the overall extinction. As an example, 3 lognormal distributions ($r_m$=100 nm, $\sigma$ =[1.2, 1.5, 1.8]) are presented in Fig. 1. Here, it is observed that the distribution is Gaussian in log space and that as the radius gets farther from $r_m$ the corresponding probability decreases rapidly. What is not obvious from Fig. 1 is that larger particles, despite having a probability equal to the smaller particles, have a disproportionate

| $\lambda$ (nm) | Sulfuric Acid (65%) | Sulfuric Acid (70%) | Sulfuric Acid (75%) | Sulfuric Acid (80%) | BC | BrC |
|---|---|---|---|---|---|---|
| 384 | 1.436 + 0i | 1.442 + 0i | 1.448 + 0i | 1.454 + 0i | 1.75 + 0.50i | 1.55 + 1.0E-2i |
| 448 | 1.421 + 0i | 1.427 + 0i | 1.433 + 0i | 1.439 + 0i | 1.75 + 0.50i | 1.55 + 4.4E-3i |
| 520 | 1.418 + 0i | 1.425 + 0i | 1.431 + 0i | 1.436 + 0i | 1.75 + 0.50i | 1.55 + 2.6E-3i |
| 755 | 1.413 + 8.66E-8i | 1.420 + 8.12E-8i | 1.427 + 7.59E-8i | 1.431 + 5.55E-8i | 1.75 + 0.65i | 1.55 + 2.0E-3i |
| 869 | 1.412 + 2.01E-7i | 1.418 + 1.96E-7i | 1.425 + 1.92E-7i | 1.428 + 1.74E-7i | 1.75 + 0.65i | 1.55 + 2.0E-3i |
| 1021 | 1.408 + 1.55E-6i | 1.415 + 1.54E-6i | 1.421 + 1.52E-6i | 1.424 + 1.44E-6i | 1.75 + 0.75i | 1.55 + 2.0E-3i |
| 1543 | 1.391 + 1.54E-4i | 1.397 + 1.48E-4i | 1.403 + 1.42E-4i | 1.405 + 1.33E-4i | 1.75 + 0.90i | 1.55 + 2.0E-3i |

**Table 1.** Complex refractive indices for sulfuric acid and smoke used in the Mie calculations. The smoke refractive index values were based on data reported in Bergstrom et al. (2002) for BC and Sumlin et al. (2018) for BrC. Sulfuric acid refractive index values are from Palmer and Williams (1975).

influence on extinction. The reason behind this disparity is the competing extinction efficiencies as shown in Fig. 2 wherein it is observed that large particles (e.g., 1 $\mu$m) are $\approx$3 orders of magnitude more efficient at attenuating light than a 70 nm particle. Logically, one can infer that one large particle can attenuate light as efficiently as 100 – 1000 smaller particles; therefore, large 120 particles, however rare, cannot be ignored when building the LUTs.

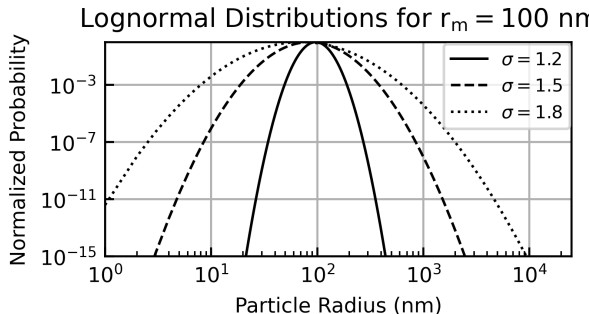

**Figure 1.** Example lognormal distributions for 3 distribution widths ($\sigma$ =[1.2, 1.5, 1.8]) for a mode radius of 100 nm.

As a further demonstration of the importance of large particles we carried out a series of simulations to demonstrate the impact that changing the upper integration boundary ($r_{max}$) in Eq. 1 has on $k(\lambda)$. Here, $\mathbf{Q_{ext}}$ was calculated for all radii between 10 nm and 10 $\mu$m (1 nm resolution) followed by the integration of Eq. 1 using different upper integration boundaries ($\mathbf{k}(\lambda, \mathbf{r_{max}})$). This model was carried out using 3 different mode radii ($r_m$=[70, 150, 500] nm) and a single distribution width 125 ($\sigma$ =1.5). Extinction coefficients were calculated as a function of the upper integration boundary of Eq. 1 ($\mathbf{k}(\lambda, \mathbf{r_{max}})$) and we calculated the percent difference between these values and the value of $\mathbf{k}(\lambda)$ when $r_{max}$ =10 $\mu$m (i.e., $\mathbf{k}(\lambda, 10\mu m)$). The results of this model are presented in Fig. 3 wherein it is observed that if the desired accuracy for $\mathbf{k}(\lambda)$ is 1% for all channels then $r_{max}$ must be $> 525$ nm when $r_m$=70 nm, $> 1\mu$m for $r_m$=150 nm, and $>2.4$ $\mu$m when $r_m$=500 nm). While changes in $\sigma$

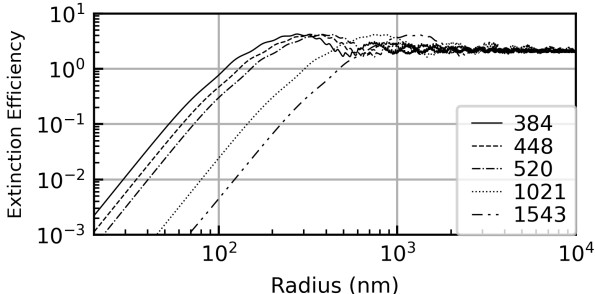

**Figure 2.** Extinction efficiencies as a function of particle size for select SAGE aerosol wavelengths.

and/or $r_m$ will modulate the limit of $r_{max}$ required to achieve 1% accuracy the general observation remains the same: choosing
a value of $r_{max}$ that is too small will invariably bias $\mathbf{k}(\lambda)$ as well as the inferred PSD values. Therefore, to obviate the impact
of this bias we set $r_{max}$ to 10 $\mu$m.

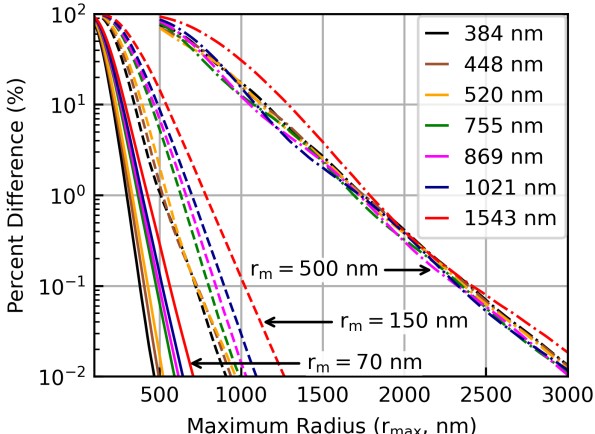

**Figure 3.** Impact of upper integration radius boundary on k. Data were referenced to $k(\lambda, 10\mu m)$. The distribution width used in this figure
was 1.5. Solid lines, dashed lines, and dot-dashed lines correspond to mode radii of 70 nm, 150 nm, and 500 nm, respectively.

### 3.1.2 Justification for resolution of distribution widths

The influence that the resolution of the distribution width ($\Delta\sigma$) had on $\mathbf{k}(\lambda)$ was evaluated as a function of $r_m$ and $\Delta\sigma$ as shown
in Fig. 4. Here, it was observed that if $\sigma =1.5$ (Fig. 4, panel b), $r_m$=75 nm (i.e., background conditions of the fine mode per
Deshler et al. (2003)), and the resolution of the LUT ($\Delta\sigma$) is 0.01 then the corresponding bias in $\mathbf{k}(\lambda)$ is $\approx$5%. It was observed
that $\sigma$ resolution has less impact as $r_m$ and $\sigma$ increased. However, in order to mitigate the bias introduced by $\Delta\sigma$ we used a
resolution of $\Delta\sigma =$1E-3 in creating the LUTs. This resolution introduces a bias in $\mathbf{k}(\lambda)$ on the order of 1% under small particle

(75 nm) conditions, which we treat as negligible in the subsequent analysis. We note that in reality size distributions are likely bimodal and the impact of the larger mode on extinction cannot be ignored as discussed below (e.g., Sec. 4.4). However, neglect of the coarse mode, with respect to the current discussion of Fig. 4, results in an over estimation of the LUT-resolution-induced error. Therefore, the percent differences in Fig. 4 are representative of upper bounds.

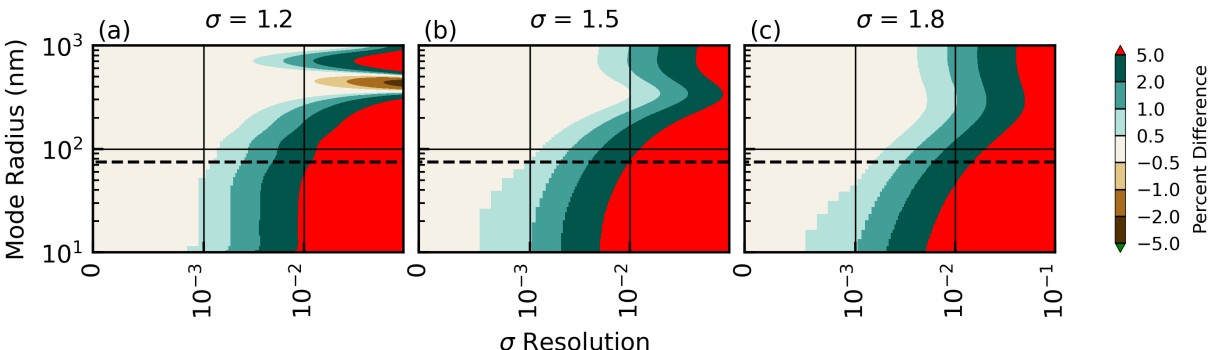

**Figure 4.** Influence of distribution width resolution on the corresponding extinction coefficient error. The wavelength used in this simulation was 520 nm. The horizontal dashed line represents the typical background particle size for the fine mode (75 nm).

Justification for the selection of the range of $\sigma$ cannot be explained with a single figure. Rather, the range of $\sigma$ was defined based on the results of a series of sensitivity tests that are described more fully in Section 4. Briefly, part of the sensitivity tests involved varying the range of PSD parameters used in LUT creation (see Table 2). It was determined that extending the LUT $\sigma$ and $r_m$ ranges to 2.0 and 1500 nm, respectively, yielded the best performing solutions when compared to theoretical data (see Sec. 4 for an expanded discussion of the corresponding sensitivity study).

| Parameter Setting # | $\sigma$ Range | $r_m$ Range (nm) |
|---|---|---|
| 0 | $1.01 - 2.0$ | $10 - 500$ |
| 1 | $1.01 - 2.0$ | $10 - 1500$ |
| 2 | $1.01 - 5.0$ | $10 - 500$ |
| 3 | $1.01 - 5.0$ | $10 - 1500$ |

**Table 2.** Range of PSD parameters used in determining the optimal range of $\sigma$ and $r_m$.

## 3.2   Inferring PSD parameters

### 3.2.1   Historical background

The necessity of using extinction ratios, as opposed to extinction coefficients, for eliminating the influence of number density (N) and inferring aerosol physical parameters has been discussed previously (e.g., Thomason et al., 1997; Bingen et al., 2006;

Wrana et al., 2021). Therefore, the method of determining the atmospheric PSD parameters is straight forward and requires only searching through the LUTs to identify all PSD parameters that yield extinction ratios that match extinction ratios from the SAGE records to within the bounds of the reported uncertainty. Indeed, if measurement error was negligible then the PSD parameters could be inferred to a high degree of accuracy. However, this level of precision is never achieved outside possibly laboratory settings and making such simplifying assumptions is fundamentally flawed. Indeed, the fundamental limitation of inferring PSD parameters from instruments like SAGE is the under-constrained nature of the problem, which results in a plethora of potential solutions.

As an illustration of the problem, LUT data were used to plot 2 extinction ratios against each other (Fig. 5, panel a). Nominal extinction ratios and their corresponding uncertainties (0.3±15%, 1.2±10%) were included as a representative SAGE data point (the red dot in Fig. 5). All LUT data that fall within the SAGE error limits are valid solutions (herein we refer to these values as "within the solution space"). Since the $r_m$ and $\sigma$ values for each LUT value is known, we can identify all $r_m$ and $\sigma$ combinations that fall within the solution space. The range and relative frequency of these values are plotted as normalized probability density functions in panels (b) and (c) of Fig. 5. Here, it is observed that, within the bounds of uncertainty, the range of $r_m$ was [160, 360] nm and the range of $\sigma$ was [1.25, 1.8]. While adding more extinction ratio combinations and decreasing the measurement uncertainty decreases the extent of the solution space, this simple example demonstrates the problem inherent in inferring PSD parameters from SAGE data: multiple solutions. Therefore, the breadth of the solution space demands the question be answered: which PSD values are either correct, most likely, or most representative of reality?

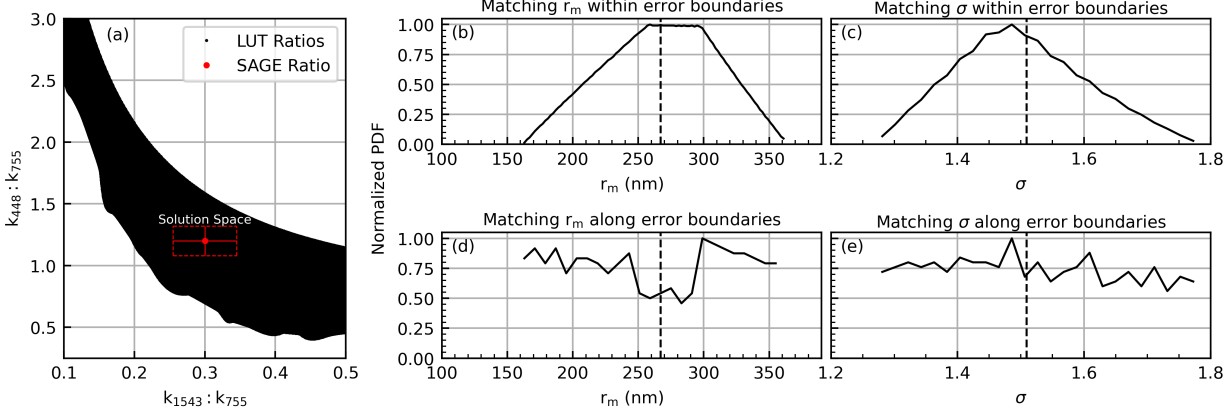

**Figure 5.** Visualization of the variability of $r_m$ and $\sigma$ within the error bars (panels b and c) and along the edges of the error limits (panels d and e). The errors were fixed at 15% and 10% along the x and y axes, respectively. The red dashed lines in (a) represent the solution space for the nominal SAGE value. The vertical dashed lines in (b–e) represent the target $r_m$ and $\sigma$ values that correspond to the red dot in (a).

A previous method published by Wrana et al. (2021) answered this by using the PSD values that fell closest to the SAGE ratios as the solution (i.e., the closest match). They went on to estimate the error in these PSD values by sampling at 8 locations along the error boundaries, as defined by the propagated errors, and calculating the mean difference between these peripheral

PSD values and those that correspond to the closest match. The challenge in this method is the highly variable nature of the PSD values that lay along these boundaries as shown in Fig. 5 (d, e). Therefore, the error estimate is highly dependent on where samples are drawn along the error boundaries.

The key points of Fig. 5 are that the central point is not necessarily a good estimate of the PSD parameters, that there is a non-negligible amount of variability in the PSD parameters within the overall solutions space, and that accurate error estimates cannot be calculated by sampling along the edge of the solution space. The method presented herein attempts to overcome these limitations.

### 3.2.2 Solutions from SAGE data

The overall flow of the solution algorithm is depicted schematically in Fig. 6. In short, extinction ratios were calculated, and uncertainties propagated, using extinction coefficients from SAGE data (for a detailed discussion of wavelength selection see Sec. 4). Corresponding extinction ratios were calculated using the LUT data. Similar to the example provided in Fig. 5 (a), the SAGE ratios were then used to find all of the LUT ratios that fell within the propagated uncertainty as well as the corresponding PSD parameters (i.e., $r_m$ and $\sigma$). While Fig. 5 (a) only shows 2 dimensions (i.e., 1 dimension per extinction ratio), the dimensionality of that figure increases as the number of extinction ratios increase, leading to a simultaneous solution in all dimensions. As discussed above, it is important to recall that finding the closest match in extinction ratios and extracting the corresponding PSD parameters is not sufficient because *all* LUT ratios that fall within the propagated measurement uncertainty are actual matches to the SAGE ratios. Therefore, a method for estimating the PSD values, given the range of values within the solution space, had to be developed. Here, we used all of the PSD values within the solution space to calculate weighted statistics (e.g., mean, median, percentiles, etc.) of the PSD parameters in an attempt to provide a statistical representation of the solution space and thereby provide a better quantification of the uncertainty in the PSD estimates.

The weights used in calculating the weighted statistics were calculated using the probability density function of a multivariate normal distribution as described by Eq. (3) where $\mathbf{x}$ is the array of LUT extinction ratios that fell within the solution space, $\mu$ is the array of SAGE extinction ratios, $\Sigma$ is the covariance matrix, and $n$ is the number of dimensions (i.e., combinations of extinction ratios) of the system. The diagonal terms in $\Sigma$ were composed of the propagated uncertainties from the SAGE data.

The construction of $\Sigma$ for Eq. (3) requires additional explanation. The covariance matrix is composed of both variances (i.e., the diagonal terms) and covariances (i.e., the off-diagonal terms that represent the degree of correlation between channels). While it is known that the variances in the SAGE extinction products are correlated, as can be can be demonstrated by calculating the coefficient of correlation between the reported errors, quantification of the covariance is not currently possible. While we cannot quantify the covariance terms we do know that the covariance will be between zero and $\sqrt{\|\boldsymbol{u}\|\|\boldsymbol{v}\|}$ (here, $\boldsymbol{u}$ & $\boldsymbol{v}$ represent the uncertainties in each channel) per the Cauchy-Schwartz inequality as shown in Eq. (4). While inclusion of the covariance terms did not significantly change the results, a good-faith estimate of covariance was attempted by setting them to mid-range values of $\frac{1}{2} \cdot \sqrt{\|\boldsymbol{u}\|\|\boldsymbol{v}\|}$.

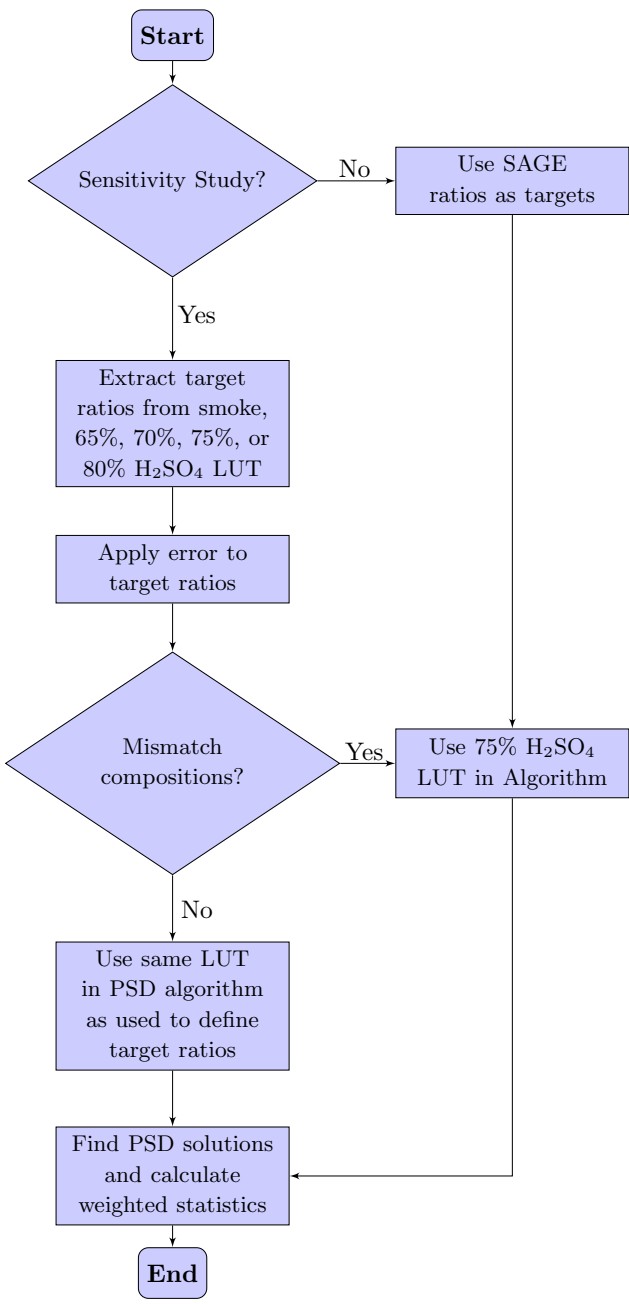

**Figure 6.** Chart describing the overall algorithm flow.

$$p(\mathbf{x}, \mu, \mathbf{\Sigma}) = \frac{\exp\left(-\frac{1}{2}(\mathbf{x} - \mu)^{\mathrm{T}} \mathbf{\Sigma}^{-1}(\mathbf{x} - \mu)\right)}{\sqrt{(2\pi)^n \det(\mathbf{\Sigma})}} \tag{3}$$

$$|\langle \boldsymbol{u}, \boldsymbol{v} \rangle| \leq \|\boldsymbol{u}\| \|\boldsymbol{v}\| \tag{4}$$

### 3.2.3  Calculation of N and higher-moment parameters

The PSD parameters that fell within the solution space were used to calculate higher-moment parameters listed in Table 3. The generalized moment equation for a lognormal distribution is shown in Eq. (5) where $\mathrm{m}_i$ is the $i$-th raw moment and N is the particle number density. Per Eq. (5), these calculations require knowledge of N. Following the method of Wrana et al. (2021), we used the PSD parameters within the solution space to calculate corresponding extinction coefficients at 1020 nm (assuming a number density of 1 cm$^{-3}$) and divided the measured $\mathrm{k}_{1020}$ value by these calculated values. As can be readily seen from Eqs. (1) and (2), this ratio results in N. Finally, all of the N, $\mathrm{r}_{\mathrm{m}}$, and $\sigma$ values within the solution space were evaluated to calculate the parameters listed in Table 3 followed by calculation of weighted statistics for these parameters as described above.

$$\mathrm{m}_i = \mathrm{N} \cdot \mathrm{r}_{\mathrm{m}}^i \cdot \exp\left(\frac{i^2 \ln^2 \sigma}{2}\right) \tag{5}$$

| Name | Symbol | Units | Equation |
|---|---|---|---|
| Number Density | N | cm$^{-3}$ | $\mathrm{m}_0$ |
| Surface Area Density | SAD | $\mu\mathrm{m}^2\mathrm{cm}^{-3}$ | $4\pi\mathrm{m}_2$ |
| Volume Density | VD | $\mu\mathrm{m}^3\mathrm{cm}^{-3}$ | $\frac{4}{3}\pi\mathrm{m}_3$ |
| Effective Radius | $\mathrm{r}_{\mathrm{e}}$ | nm | $\frac{\mathrm{m}_3}{\mathrm{m}_2}$ |

**Table 3.** Description of higher-moment parameters that were calculated using the inferred PSD parameters and Eq. (5).

## 4  Sensitivity Tests

### 4.1  Composition is correctly assumed

Before applying this method to the SAGE records we sought to evaluate the accuracy and consistency of this method for inferring PSD values. This was achieved, from a theoretical perspective, by using LUT data and involved 4 overall steps:

1. Select extinction coefficients from the LUT. These coefficients, which have known PSD parameters, act as the pseudo-SAGE data.

2. Impose a nominal uncertainty on the pseudo-SAGE extinction coefficients, calculate the extinction ratios and propagate the errors. Here, the errors were held constant across all wavelengths and the evaluation was repeated for errors that ranged from 1% through 50%.

     3. Run the pseudo-SAGE extinction ratios through the solution algorithm to calculate the inferred PSD statistics. Here, the algorithm uses the same LUT from which the pseudo-SAGE extinction coefficients were pulled in step 1.

4. The inferred PSD parameters were compared to the input values to determine how well the two matched.

The overall flow of the algorithm is depicted schematically in Fig. 6. This evaluation was repeated for a series of extinction ratio combinations (see Table 4) and LUT boundary conditions (Table 2) to determine which combination yielded the most accurate results.

Based on the results of this evaluation we determined that condition #5 (Table 4) and particle parameter setting #1 (Table 230 2) yielded the best overall performance. This is not surprising as this condition used all of the available channels, which takes advantage of all of the information content available in the SAGE data. The results of this simulation are seen in Fig. 7 where the inferred-to-target ratios were plotted for 3 different error values (i.e., 50%, 20%, and 5% error) as a function of the inferred PSD value. The red horizontal lines in Fig. 7 indicate median values, the boxes extend from $P_{25}$ to $P_{75}$ and the whiskers indicate $P_5$ and $P_{95}$. Here, we chose to plot the inferred-to-target ratio as a function of the inferred PSD parameter because this is most 235 applicable to real-world use cases (i.e., where the "real", or target value is unknown). For example, if the reported errors of the extinction coefficients were all within 5% and the inferred mode radius was 100 nm then, per Fig. 7 (c), we know that on median the inferred mode radius is $\approx$5% too high, and that 90% of the time the inferred value is within $\pm$15% of the target value.

While the performance of condition #5 was optimal it can significantly reduce the number of SAGE extinction spectra that 240 yield a viable PSD inference (see Fig. 8 panels (a) and (d)). This is caused by 2 related issues: 1. shorter-wavelength channels attenuate higher in the atmosphere and result in extinction ratios that fall outside the LUTs' range, 2. using so many channels increased the chance of having a negative extinction coefficient (Kovilakam et al., 2023) or an otherwise invalid value (e.g., set to "fill" values). A similar combination of extinction ratios that excluded the 384 nm channel was evaluated (condition #6). While the performance of condition #6 was comparable to condition #5, there was a noticeable decrease in accuracy (not 245 shown). Further, excluding the 384 nm channel alone did not significantly increase the data volume as shown in Fig. 8. While excluding the 384 nm channel yielded more solutions at lower altitudes, the failure rate remained >30% at mid-latitudes (Fig. 8 panels (b) and (e)). As a final test we evaluated the performance of a condition definition that used only 3 channels (#15, panels (c) and (f)). This wavelength combination resulted in valid PSD estimates for more than 90% of its data at all altitudes and latitudes. Figure 8 also shows the performance of these 3 conditions when only the highest-quality SAGE data were used 250 (i.e., when the reported error was $\leq$20%; panels d–f), which did not significantly change the data throughput.

The wavelength combinations used by Wrana et al. (2021) (condition #15 in Table 4) had the third-best overall performance (see Fig. 9) and have the added benefit of only relying on 3 extinction channels, which resulted in a major improvement in data retention (see Fig. 8 panels c and f). While this condition lacked the overall accuracy of conditions #5 & 6, we determined that

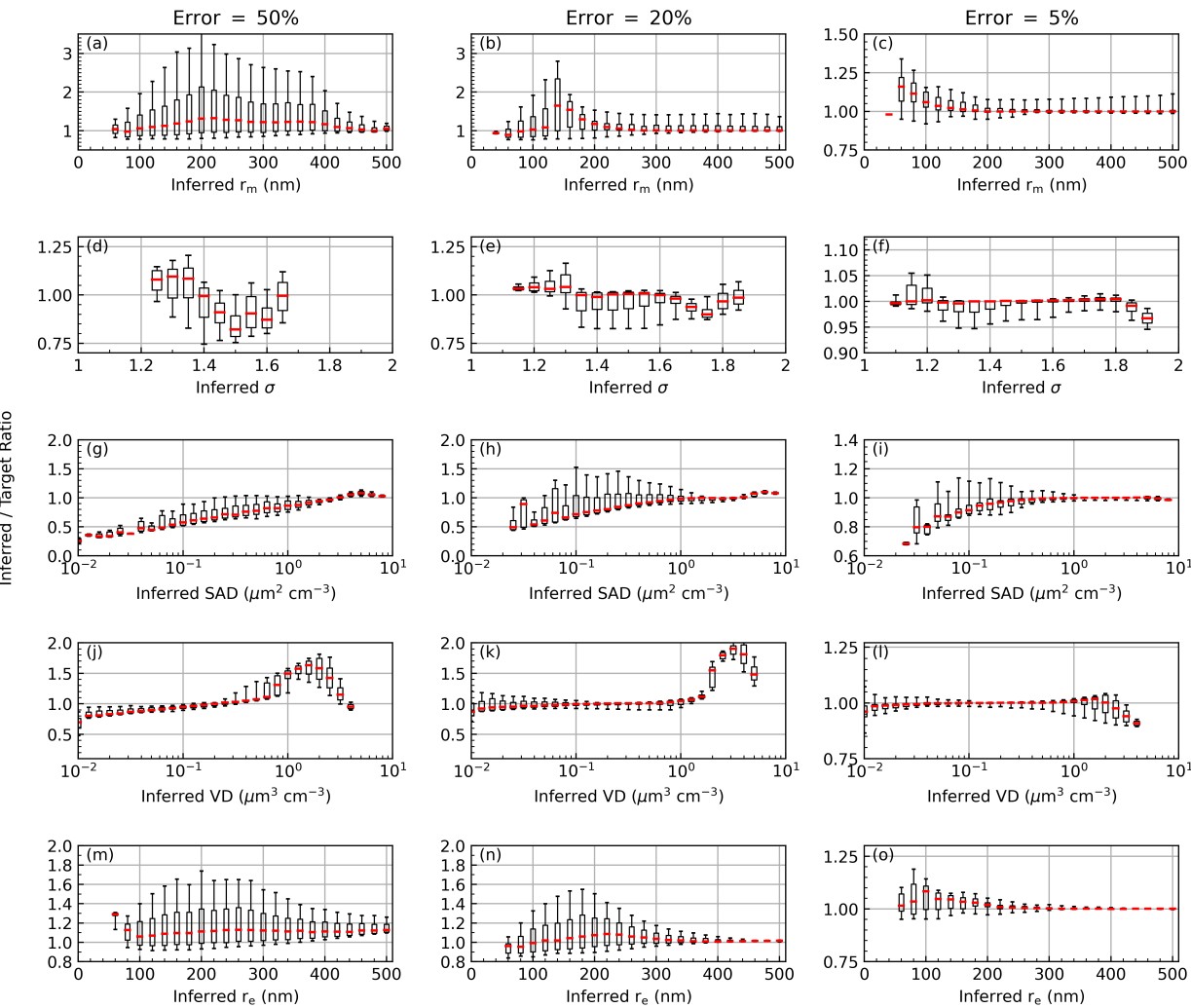

**Figure 7.** Results of the sensitivity study for condition #5 (Table 4) and particle parameter setting #1 (Table 2) at 3 different error values. The red lines indicate median values, the boxes extend from $P_{25}$ through $P_{75}$, and the whiskers represent $P_5$ and $P_{95}$.

| Condition # | Numerator Wavelengths (nm) | Denominator Wavelength (nm) |
|---|---|---|
| 0 | 520 | 1021 |
| 1 | 448, 520 | 1021 |
| 2 | 448, 520, 755 | 1021 |
| 3 | 448, 520, 755, 869 | 1021 |
| 4 | 384, 448, 520, 755, 869 | 1021 |
| 5 | 384, 448, 520, 755, 869, 1543 | 1021 |
| 6 | 448, 520, 755, 869, 1543 | 1021 |
| 7 | 520 | 1543 |
| 8 | 448, 520 | 1543 |
| 9 | 448, 520, 755 | 1543 |
| 10 | 448, 520, 755, 869 | 1543 |
| 11 | 448, 520, 755, 869, 1021 | 1543 |
| 12 | 520, 755, 869, 1021 | 1543 |
| 13 | 384, 448, 520, 755, 869 | 1543 |
| 14 | 384, 448, 520, 755, 869, 1021 | 1543 |
| 15[*] | 448, 1543 | 755 |

**Table 4.** Condition definitions and wavelength assignments for extinction coefficient ratios used in the current study. The asterisk indicates the wavelength combination used in Wrana et al. (2021).

its performance was acceptable for use as an alternate within this method. Therefore, the results of this evaluation indicate that
if we work strictly within the confines of theory then condition #5 is the optimal choice. However, application of this method
to reality (i.e., where we must account for measurement uncertainty, the possibility of saturating channels, etc.) we conclude
that condition #15 is the optimal choice. This puts us on the horns of a dilemma where we are forced to choose our concession:
do we sacrifice latitude/altitude coverage for improved accuracy (condition #5) or sacrifice accuracy for improved coverage
(condition #15)? While further discussion on the resolution of this dilemma is outside the scope of this section we will state
that a hybrid model that involves a combination of conditions #5, #6, and #15 were used in the application to the SAGE data
and the reader is directed to Section 6 for further discussion.

In summary, the general observation of Figs. 7 & 9 is that the accuracy of the PSD solutions became better as the measurement uncertainty decreased and as the input PSD parameters become more like an enhanced event (e.g., larger $r_m$, enhanced
SAD, etc.). While this is encouraging, it must be recognized that this simulation is only theoretical in nature and that the
more challenging aspects of real-world aerosol compositions (e.g., mixed composition particles, multi-modal distributions,

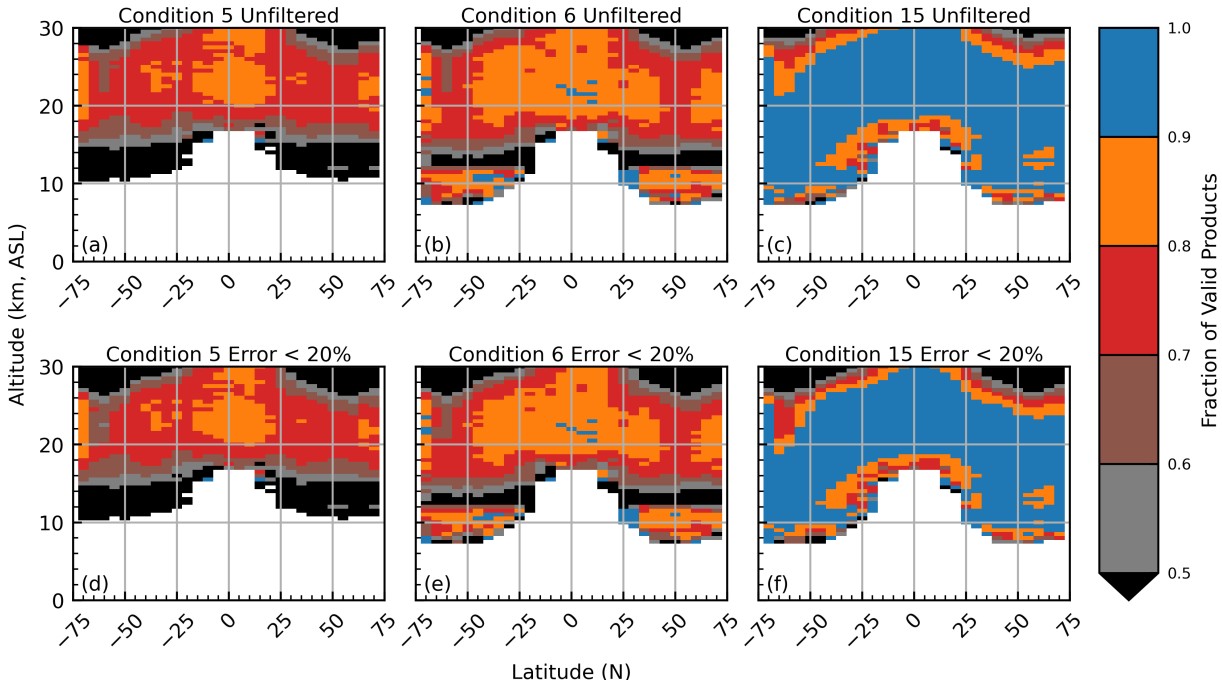

**Figure 8.** Zonal representation of the fraction of SAGE data that yielded PSD estimates. Panels d-f display the statistic when data with errors >20% were removed. The fraction was calculated by dividing the number of successful PSD estimates by the total number of SAGE extinction spectra within a given altitude and latitude bin.

etc.) and PSD parameters have been neglected. To help address these limitations we evaluated the impact of assuming a wrong composition below.

## 4.2 Impact of assuming a wrong sulfuric acid content

To this point we have neglected to account for varying aerosol compositions and how making the incorrect assumption about the composition may influence the inferred PSD parameters. In this section we present the results of a simulation wherein we evaluated the impact of assuming the wrong weight percent $H_2SO_4$. Because $H_2SO_4$ is typically assumed to be 75% we use that as the point of reference.

Figure 10 demonstrates the overall impact of assuming an incorrect weight percent of $H_2SO_4$ in the PSD algorithm. Here, the algorithm searched through the 75% $H_2SO_4$ LUT to find all extinction ratios within the solution space whereas the pseudo-SAGE extinction ratios were pulled from the 65%, 70%, and 80% $H_2SO_4$ LUTs (see Fig. 6 for workflow and Table 1 for refractive indices).

The y-axes in Fig. 10 represent the ratio of the inferred PSD parameter when the LUTs were mismatched to the inferred PSD parameters when the LUTs were correctly matched (i.e., both the extinction ratio and algorithm LUTs were 75% $H_2SO_4$).

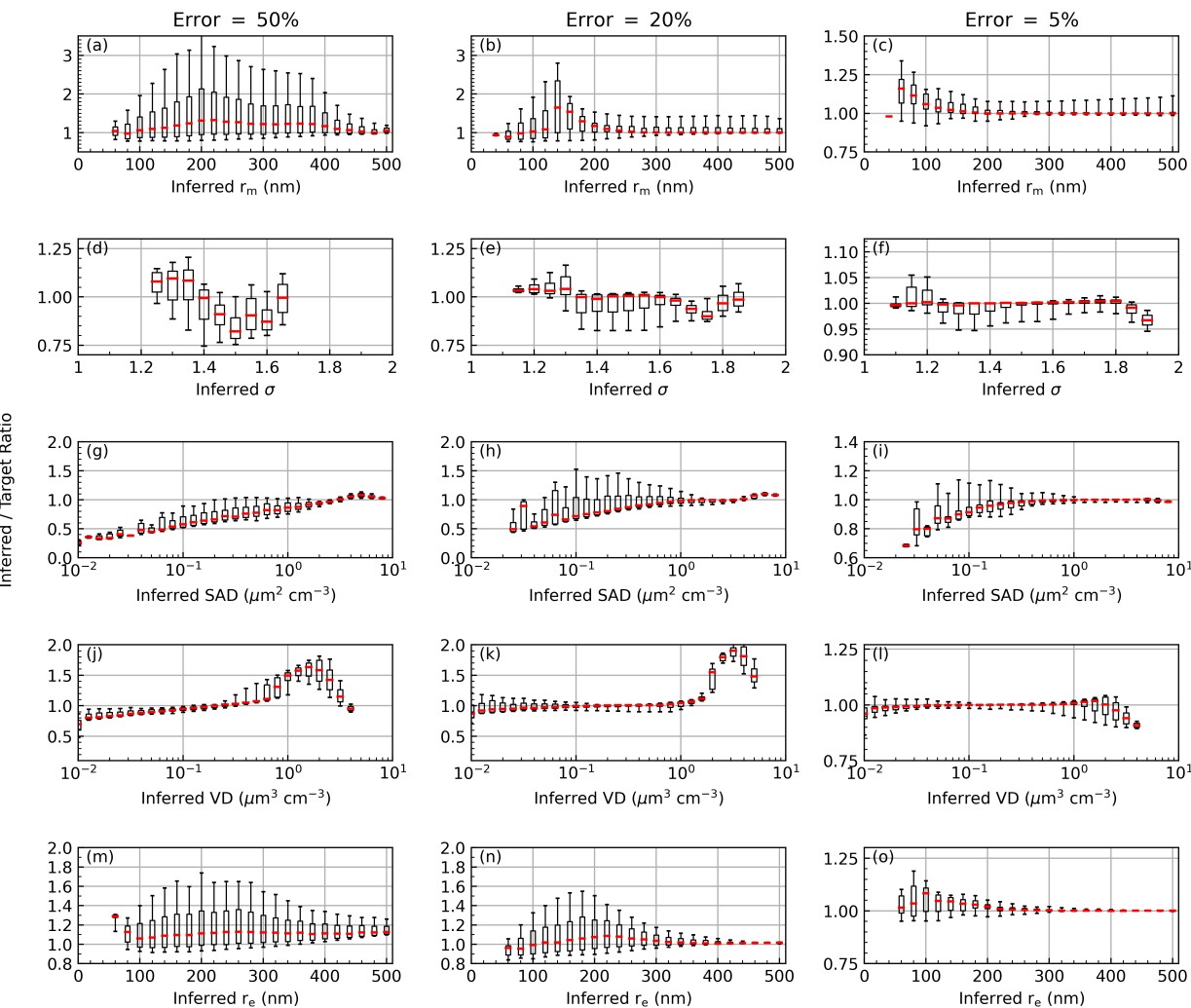

**Figure 9.** Same as Fig. 7, but for condition #15.

What this tells us is how much an incorrect assumption about the $H_2SO_4$ weight percent changes the inferred PSD parameters as compared to Figs. 7 & 9 (i.e., how much worse are these estimates as compared to getting the composition correct). The extinction coefficient errors used to create Fig. 10 were set to 20%. Indeed, the spread in the inferred PSD parameters narrowed with decreasing error and the 20% error solutions are presented as a representative example.

The general observation from Fig. 10 is that making incorrect assumptions about the aerosols' $H_2SO_4$ content has minimal impact on the inferred distribution widths (all were within $\approx$1–2% as compared to Fig. 7) and the inferred $r_m$ was most impacted (within $\pm$5%). These deviations were compounded in the inferred SAD and volume density (VD) products, which deviated by <8%.

We note that overall the influence of incorrect $H_2SO_4$ assumptions was consistent across all condition definitions (Table 4) with only minor variations. We conclude that incorrect assumptions about the $H_2SO_4$ content had minor impact on the accuracy of the inferred PSD parameters (generally within $\pm$5%). Since the $H_2SO_4$ content of atmospheric aerosols is, ultimately, unknown we note that this situation adds an unknown element to the analysis. However, this uncertainty can be partially mitigated by using concurrent water vapor and temperature observations to estimate $H_2SO_4$ content at thermodynamic equilibrium (Steele and Hamill, 1981; Bernath et al., 2023) and will be a topic of study for future releases of this product.

### 4.3   Impact of assuming a smoke-free atmosphere when smoke is present

The performance of the PSD algorithm was evaluated to determine its accuracy in estimating PSD parameters when the atmosphere is assumed to consist solely of 75% $H_2SO_4$ when it really has smoke. As stated above, smoke is challenging to model because of the ambiguity in its composition and physical properties (e.g., refractive index). While recent work indicates that stratospheric smoke may be consistent in composition and size distribution (Thomason and Knepp, 2023) there have been no in situ observations to determine the composition and physical properties of stratospheric smoke. Therefore, we conducted the following simulation under the assumption that most smoke is composed of brown carbon (BrC) with little contribution from black carbon (BC) and used the refractive indices of Sumlin et al. (2018) and Bergstrom et al. (2002) for BrC and BC, respectively (refractive index values in Table 1).

This simulation followed the same methodology as in Sec. 4.2, but used the smoke LUTs as the extinction ratio targets (see flow in Fig. 6). Here, we assumed that 90% of the particles were composed of 75% $H_2SO_4$ with $r_m$=75 nm, and $\sigma$=1.5 and that 10% of the particles were composed of smoke of various compositions. The labels in Fig. 11 indicate the relative breakdown of the smoke particles between BrC and BC.

While $\sigma$ was consistently high the deviation in $r_m$ was minimized when the inferred $r_m$ was between 100 and 400 nm. This falls in an ideal location as previous studies have shown that pyroCb-related smoke particles are typically between 125 and 250 nm (Moore et al., 2021; Katich et al., 2023). While these studies were conducted in the troposphere (Katich et al. (2023) sampled near the Arctic lower stratosphere) we take them to be representative of stratospheric values for this study, though we recognize that additional in situ sampling would be highly beneficial. Finally, we note that both SAD and VD were significantly underestimated under reasonable values (e.g. when SAD $\geq$0.1 $\mu$m$^2$ cm$^{-3}$), but these discrepancies effectively canceled out in

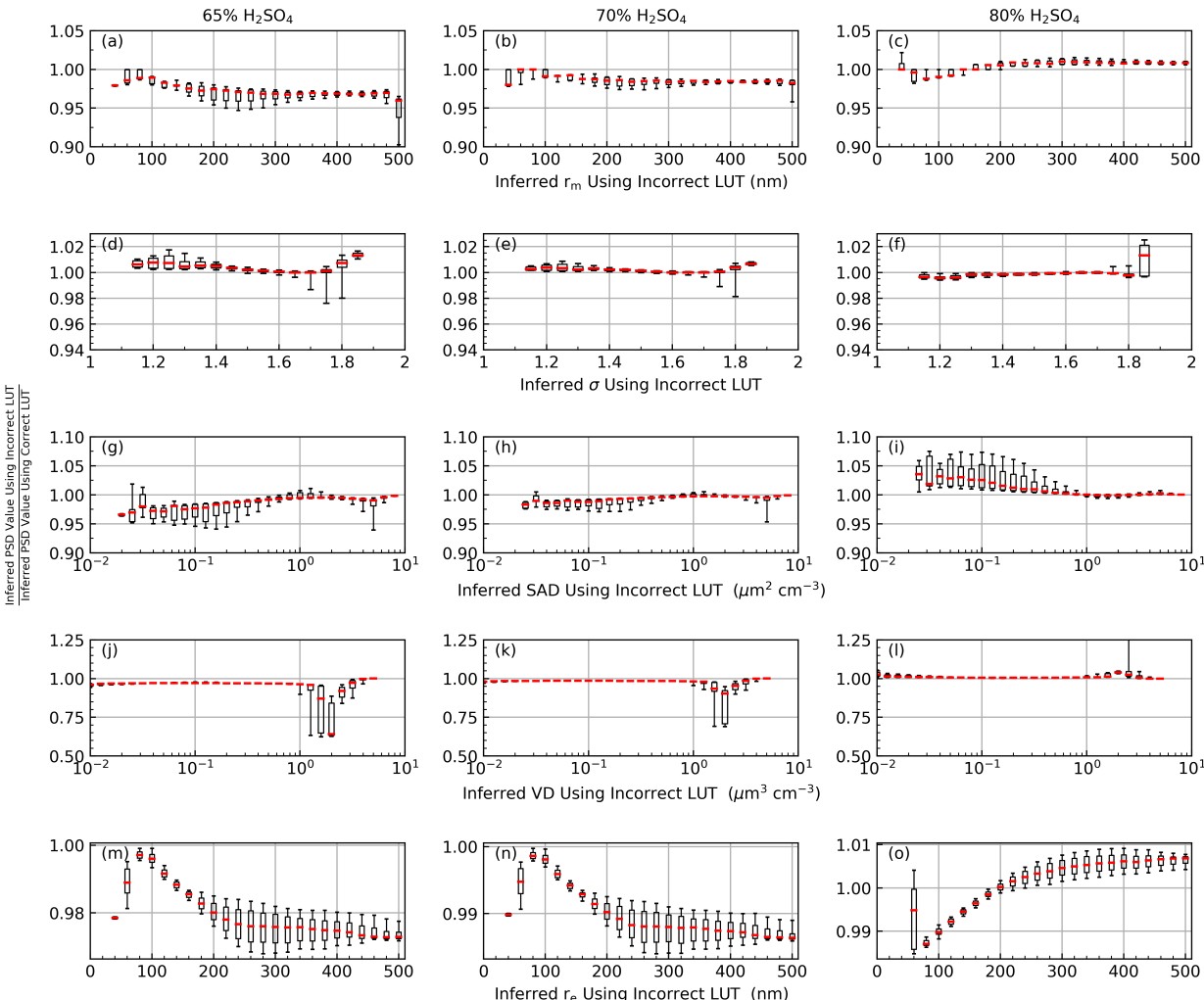

**Figure 10.** Visualization of the impact of getting the weight percent $H_2SO_4$ incorrect. The presented data came from condition #5 with an imposed extinction coefficient error of $\pm20\%$.

the calculation of $r_e$. Given these considerations, we must provide a cautionary note when using data that may be contaminated by the presence of smoke.

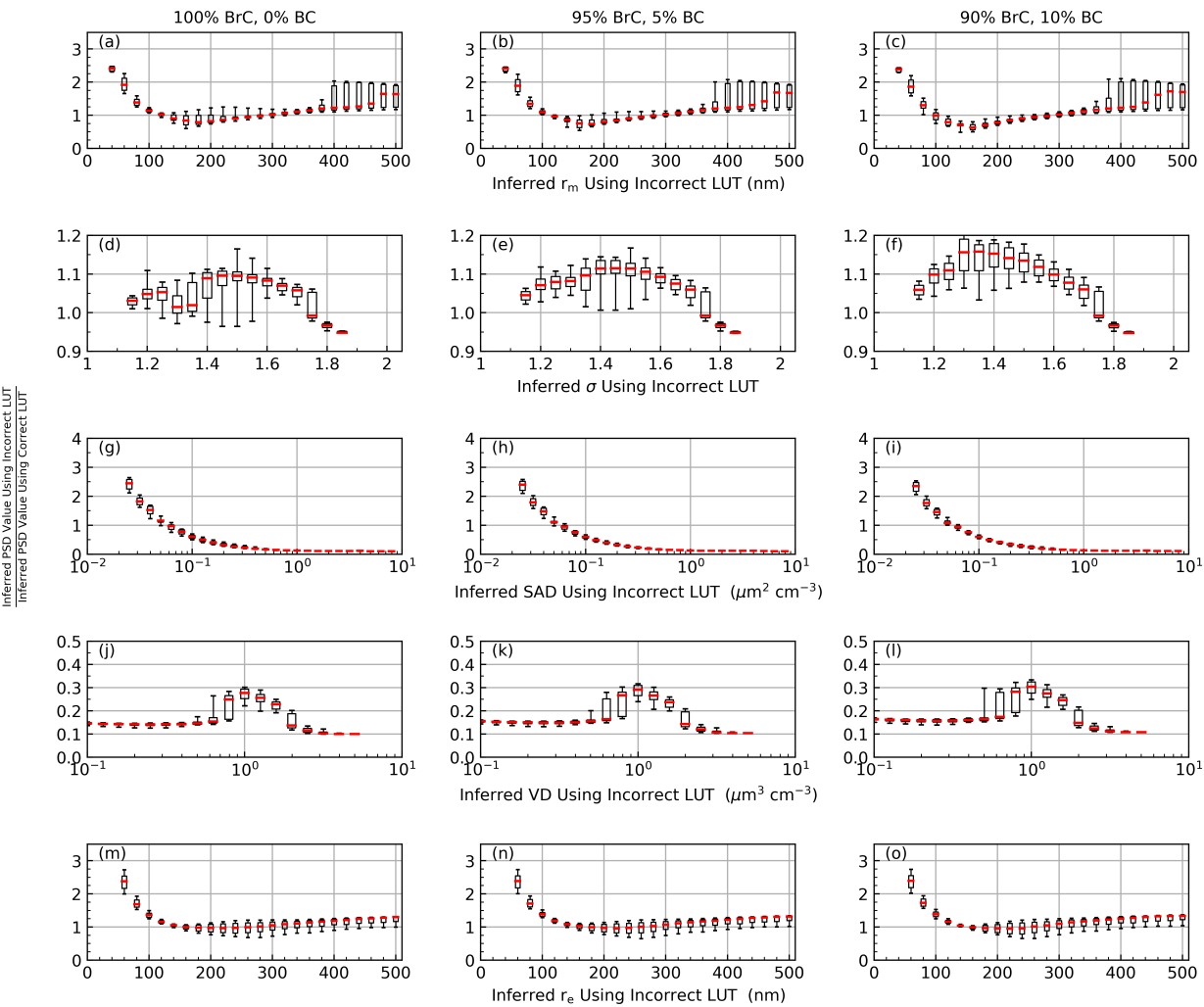

**Figure 11.** Visualization of the impact of assuming a smoke-free atmosphere when smoke is present. The presented data came from condition #15 with an assumed extinction coefficient error of $\pm 20\%$.

## 4.4 Influence of a second mode: an OPC case study

Expanding the solution space from single-mode to bimodal distributions greatly expands the number of possible solutions (see Section 5 for additional discussion) and expands the number of variables to solve for. This makes visualization and interpretation of the results challenging. Therefore, we limit this section to a higher-level demonstration that provides the reader with a general understanding of the implications of incorrectly assuming single-mode distributions. To aid in this we

used the University of Wyoming's Optical Particle Counter (UWY OPC) record, which provides bimodal PSD parameters at 0.5 km resolution (Deshler, 2023). This record consists of 150 profiles that were collected over Laramie, WY (41.3 °N, 105.58 °W) between 1989–2019 and include observations in the wake of major eruptions such as Mt. Pinatubo (1991) and Raikoke (2019). The OPC PSD parameters were used to calculate extinction coefficients at SAGE wavelengths (see Eq. (1)) under 2 conditions: 1. using only the first mode, 2. using both modes. These OPC-based extinction coefficients were used to create pseudo-SAGE extinction ratios, which were then fed into the PSD solution algorithm, which still used a single-mode lognormal distribution LUT. The uncertainty of the extinction coefficients was fixed to a highly conservative value of 5% (Deshler et al., 2003). Herein the aerosol composition was set to 75% $H_2SO_4$.

### 4.4.1 Only using the first OPC mode to calculate extinction

The OPC-based extinction coefficients were calculated using only the first mode in the OPC data followed by running these extinction coefficients through the PSD inference algorithm. This effectively tests the algorithm's performance when the number of modes in the atmospheric aerosol distribution matches the number of modes in the LUT.

Figure 12 (panels a–f) demonstrates how the inferred PSD parameters and microphysical properties compared to those reported in the OPC record. Here, the inferred values were divided by the reported OPC values for comparison and the shaded region represents the median $\pm1.4826$ times the median absolute deviation (MAD[*]). The median and MAD statistics are statistically robust alternatives to the mean and standard deviation and are less susceptible to impact from outliers (Leys et al., 2013). Further, when MAD is multiplied by 1.4826 it is roughly equivalent to 1 standard deviation (Leys et al., 2013).

It was observed that the inferred $r_m$ was consistently overestimated by up to $\approx$30% (on median), though the agreement became better towards the middle stratosphere (20–25 km). The distribution width provided the best agreement, while the SAD and VD were consistently underestimated by $\approx$25%. This resulted in an underestimation of effective radius ($r_e$) on the order of $\approx$10–20%. These results are consistent with the initial single-mode evaluation (Figs. 7 and 9). Therefore, we conclude that if the atmosphere's aerosol is distributed within a single-mode lognormal distribution and the LUT is likewise single-mode lognormal then the performance remained consistent with the previous theoretical evaluations. The remaining question is: how does the performance change if the atmosphere's aerosol distribution is bimodal?

### 4.4.2 Using both OPC modes to calculate extinction

The analysis was repeated using both OPC modes to calculate the pseudo-SAGE extinction coefficients that were fed into the PSD inference algorithm. We reiterate that the solution algorithm is still searching for solutions within single-mode lognormal LUTs. This tests the viability of inferring representative single mode PSD parameters when the atmosphere has aerosol that has a bimodal distribution.

The results of this analysis are seen in panels g–l of Fig. 12 wherein a stark change from the results in panels a–f is observed. Here, the inferred $r_m$, $\sigma$, and N were referenced to the first mode values reported in the OPC record. It was observed that addition of the second mode significantly increased the inferred $r_m$, $\sigma$, and VD. Overall, this also increased $r_e$, especially at low

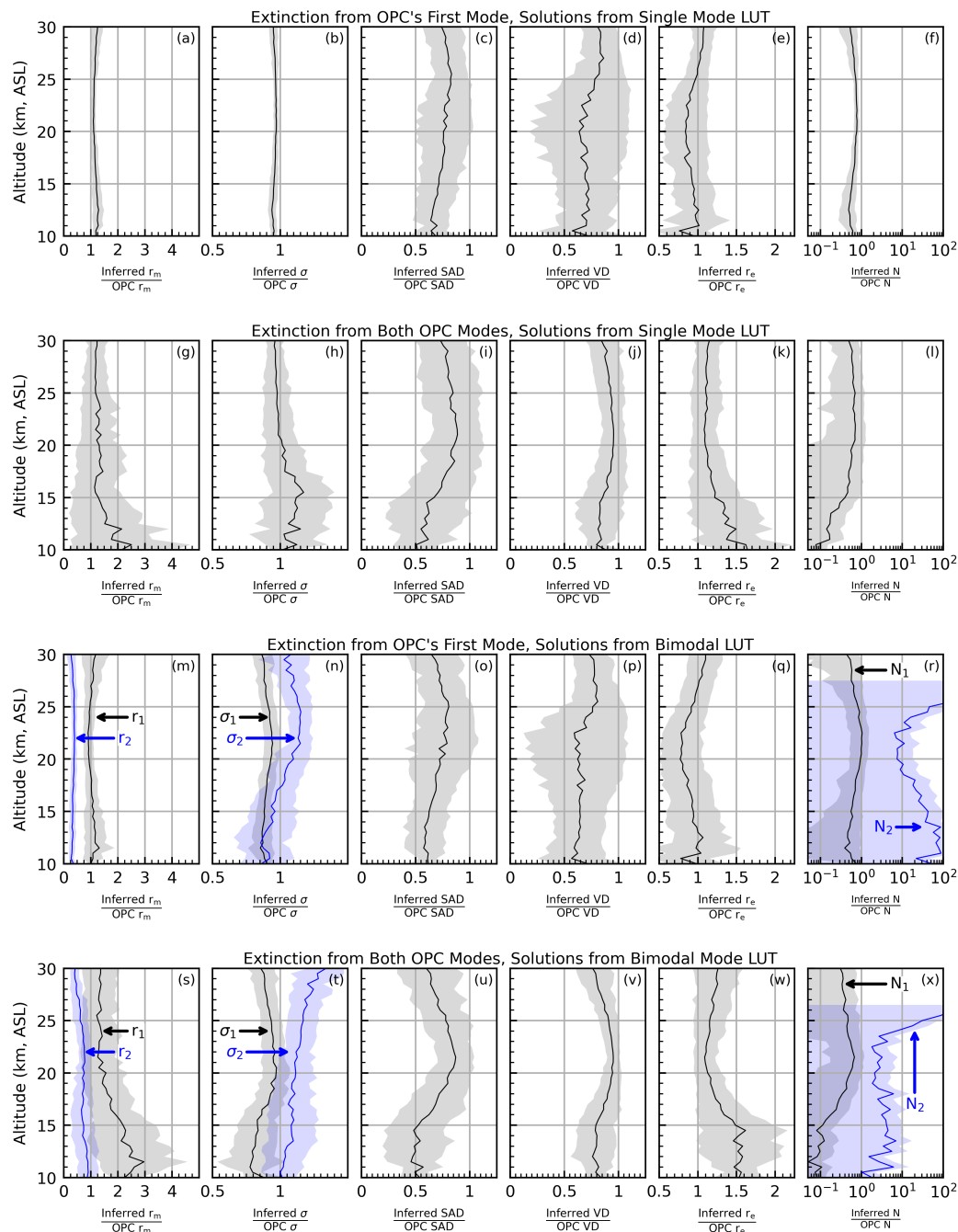

**Figure 12.** Solution profiles of the inferred PSD parameters and microphysical properties referenced to the reported OPC value. The solid lines represent the median ratio and the shaded regions represent the median $\pm\mathrm{MAD}^*$. The uncertainty imposed on the extinction coefficients was fixed at 5%.

altitudes, though the middle-stratosphere performance remained comparable to that observed in panel (e). Finally, addition of the second mode reduced the inferred number density by up to a factor of 10 at low altitude (panels f, l).

While a change in performance was not unexpected, the degree of influence the second mode had on the inferred PSD parameters may not be intuitive, especially when one considers the relative paucity of larger-mode particles (OPC number density ratio profiles are shown in panel (d) of Fig. 13 for reference; see also additional discussion below). To understand this one must recall the disproportionate influence large particles have on the overall extinction as shown in Fig. 2. This is further illustrated in Fig. 13 where panel (a) shows the second mode's contribution to the overall extinction as a function of the second mode's $r_m$ (labeled $r_2$) for 3 different wavelengths (385, 520, 1020 nm) and 2 number density ratios ($N_2:N_1 = $ 1E-2 and 1E-3). Panel (b) of Fig. 13 shows the second mode's contribution to overall extinction as a function of number density ratios for the same 3 wavelengths with the following PSD parameters: $r_1$=75 nm, $\sigma_1$=1.45, $r_2$=310 nm, $\sigma_2$=1.05 (median values from the OPC record). Panels (c) and (d) aid in interpreting panels (a) and (b) by presenting quartile profiles for $r_2$ and the number density ratio as calculated from the OPC record.

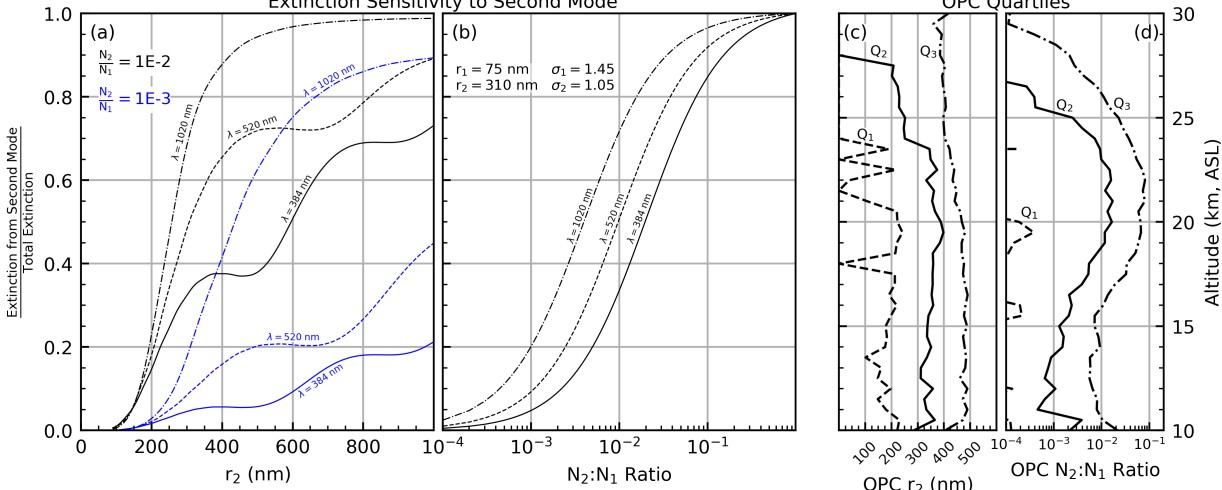

**Figure 13.** Fraction of the total extinction that comes from the larger mode. Panel (a) plots this fraction as a function of the second mode radius ($r_2$) for 3 wavelengths and 2 different number density ratios. Panel (b) plots this fraction as a function of number ratio for 3 wavelengths using constant bimodal PSD parameters (text inset within the figure). Panels (c) and (d) present the quartile profiles, from the OPC record (see text for details), for the $r_2$ and number density ratio, respectively.

What is first observed in Fig. 13 (a) is a rapid increase in the second mode's contribution to the overall extinction as particle size increases, particularly at the longer wavelengths. However, the shorter wavelengths show an interesting behavior in that their extinction increased rapidly followed by a flattening and slight decrease in extinction, which is subsequently followed by another rapid increase (this is a product of resonances in the Mie scattering, notably when $r_2$ equalled an integer multiple of $\lambda$). Indeed, the longer wavelength data demonstrate this same behavior, though not on the scale of this figure. The cause of this

flattening is a combination of the extinction efficiency for the specified particle as well as the width of the lognormal distribution that was applied prior to integration. For example, if $r_m$ falls near the peak in the efficiency curve and the distribution width is sufficiently narrow then the integrated efficiency (or extinction) will be effectively constant as $r_m$ changes. Therefore, where this flattening occurs on the x-axis of Fig. 13 (a) depends not only on $\sigma$ but also on the wavelength and how the number densities are allocated between the 2 modes.

This figure also demonstrates that at middle stratospheric conditions (20-25 km) where $r_2 \approx 350$ nm (panel (c)) and $N_2:N_1 \approx 1E$-2 (panel (d)), the second mode accounts for between $\approx 35\%$ ($\lambda = 384$ nm) and $\approx 85\%$ ($\lambda = 1020$ nm) of the overall extinction. Further, panel (d) shows that, in the OPC record, $N_2:N_1$ varied between 1E-3 and 1E-2 from $\approx 14$–25 km. This variation in number density results in the second mode contributing between 5% ($\lambda = 384$ and $N_2:N_1 = 1E$-3) and $\approx 75\%$ ($\lambda = 1020$ and $N_2:N_1 = 1E$-2) of the overall extinction. Therefore, this simulation demonstrates how sensitive the SAGE extinction spectrum can be to larger-particles even when these particles are outnumbered 1000:1. This results in a inevitable bias in not only our PSD algorithm but all algorithms that use SAGE, and SAGE-like, data to infer PSD parameters. Indeed, it is for this reason that the inferred PSD parameters in panels g–l of Fig. 12 were larger than those in panels a–f. We reiterate that the degree of this bias is dependent on numerous factors and urge caution in trying to disentangle this information. Ultimately this adds an additional challenge to inferring PSD parameters from SAGE data, one that cannot be fully resolved without incorporation of additional information from other instruments.

## 5  Evaluation of the practicality of bimodal solutions

The solution algorithm discussed above was expanded to accommodate a bimodal solution space. Inclusion of the second mode is particularly intriguing in light of the discussion in Sec. 4.4.2. A complicating factor in moving to bimodal distributions is that number density cannot be completely ignored. Indeed, the dimensionality of the LUTs must be expanded to not only account for the second $r_m$ and $\sigma$ values, but also to account for the relative distribution of particles across the first and second modes. This expands the number of PSD parameters that were used to create the LUTs from 2 ($r_m$ and $\sigma$) to 6 ($r_{1,2}$, $\sigma_{1,2}$, $N_{1,2}$) as well as an overall expansion in the size of the LUTs. For example, if the bimodal LUTs were constructed using the same limits and resolutions that were used in the single-mode analysis then the LUT would expand from $\approx 41.6E6$ values ($\approx 160$ MB at 32-bit precision) to more than 1.7E15 values ($\approx 7$ PB at 32-bit precision; this number assumes a *single* ratio of $N_1$ to $N_{total}$). Therefore, the bimodal LUT limits and resolutions were reduced to accommodate these issues as shown in Table 5. The limits of these parameters were informed by the PSD parameters reported by the UWY OPC record (excluding data collected between 1-June 1991 and 1-Jun 1997 to remove the impact of the Pinatubo eruption). The OPC-based statistical profiles are presented in Fig. 14 for reference.

Similar to the single mode sensitivity study (Sec. 4) the bimodal LUTs were tested to determine the feasibility of accurately identifying all six PSD parameters ($r_{1,2}$, $\sigma_{1,2}$, $N_{1,2}$). We recognize that the limitations on the range and resolution of PSD values in these LUTs inevitably influenced the accuracy of these results (see Sections 3.1.1 & 3.1.2 for discussion). Further, we recognize that these limits do not cover the full range of PSD values reported within the UWY OPC record. Indeed, if the

| Parameter | Range/Values | Resolution |
|---|---|---|
| $r_1$ (nm) | $10 - 1000$ | 10 |
| $r_2$ (nm) | $100 - 2000$ | 10 |
| $\sigma_1$ | $1.1 - 1.8$ | 0.01 |
| $\sigma_2$ | $1.01 - 1.6$ | 0.01 |
| $(N_1/N_{total})$ | 1, 0.999, 0.99, 0.975, 0.95, 0.90 | NA |
| Extinction error (%) | 5 | NA |

**Table 5.** Range and resolution of PSD parameters used in creating the bimodal LUTs.

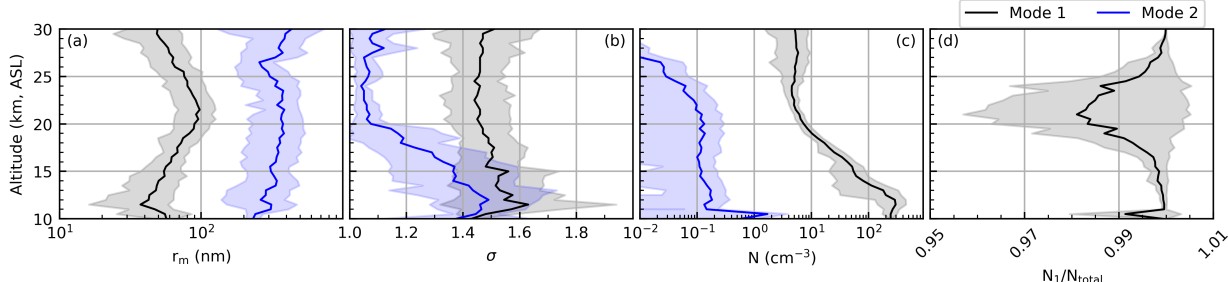

**Figure 14.** Profiles of PSD parameters as reported in the University of Wyoming's OPC record. Data collected between 1-June 1991 and 1-June 1997 were excluded to remove the influence of the 1991 eruption of Mt. Pinatubo. The shaded regions indicate the median $\pm$MAD[*].

shaded regions in Fig. 14 were expanded to cover 90% of the reported values instead of the median$\pm$MAD[*] then the range of values would have been substantially larger. Therefore, we explicitly state that, as constructed, this simulation is not designed to be applicable to the full range of possible atmospheric conditions. Rather, we defined the boundaries of this model using reasonable values that cover the majority of PSD values as reported in the UWY OPC record. Further, the intent of this bimodal test is not to determine, necessarily, how accurately the inferred bimodal PSD values are, but it is to determine the stability of operating within this expanded solution space.

## 5.1 OPC case study

The OPC record was used to evaluate the performance of the bimodal solution algorithm in a manner similar to that discussed in Sec. 4.4 and the results are shown in Fig. 12 panels (m–x). Panels (m–r) of Fig. 12 show the results when the pseudo-SAGE extinction coefficients were calculated using only the first mode PSD parameters from the OPC data (i.e., tests the performance when the atmospheric aerosol is single mode but the solution algorithm uses a bimodal LUT). The reader may question how the second-mode ratios of panels (m) and (n) were calculated if the OPC-generated extinction spectra only used the first OPC mode? Here, the inferred PSD parameters were divided by the first-mode of the OPC data. While this may seem nonsensical, this depiction is important for 2 reasons. First, the difference between the $r_1$ and $r_2$ curves in (m) (and the $\sigma_1$ and

$\sigma_2$ curves in (n)) demonstrate that the solution algorithm does not try to force both modes on top of each other (i.e., $r_1 \neq r_2$ and $\sigma_1 \neq \sigma_2$). The inferred modes are distinct from each other. Second, this shows that the inferred first-mode parameters are in better agreement with the first mode of the OPC data (i.e., the mode that was used to generate the pseudo-SAGE extinction spectra). This demonstrates that the solution algorithm does not try to force all of the spectral information into the second mode and, consequently, minimize the first mode. In general, this is a demonstration that the algorithm is performing as intended.

Finally, panels (s–x) of Fig. 12 show the results when the pseudo-SAGE extinction coefficients were calculated using both OPC modes (i.e., tests the performance when both the atmospheric aerosol and the solution algorithm LUT are bimodal).

      Here, it is observed that mismatching the number of modes in the LUT and atmospheric aerosol distribution had a modest impact on the estimate of $r_1$ and $\sigma_1$, though $r_2$ was underestimated by $\approx 90\%$. This resulted in a highly unreliable estimate of the second mode's number density ($N_2$, panel (r)). While the corresponding SAD and VD estimates were underestimated,

this resulted in an inferred $r_e$ that was within $\approx 25\%$ of the OPC value (on median) throughout the profile. Overall, this model indicates that the bimodal solution algorithm provides reasonable estimates for $r_1$ and $r_e$ despite the pseudo-SAGE extinction coefficients being built off aerosols from a single-mode distribution.

      A stark change in performance was observed when the pseudo-SAGE extinction coefficients were created using both OPC modes (panels s–x). Here, $r_1$ was consistently overestimated by $\geq 25\%$ throughout the profile while $r_2$ fell much closer to the

430 1:1 line. The overestimation of $r_1$ is in agreement with the discussion in Sec. 4.4.2. While the overestimation of $N_2$ is not as severe as in panel (r), the algorithm continued to overestimate $N_2$ by a factor of $\approx 5$ and $N_1$ is now underestimated throughout much of the profile. This translates into an overall improvement of the SAD and VD parameters and $r_e$ were within 25% above 17 km. Ultimately this demonstrates how the second mode dominates the performance of the solution algorithm.

### 5.2   Instability of bimodal solutions from a theoretical perspective

The stability of the bimodal solutions was evaluated in a manner similar to that discussed in Sec. 4. Extinction coefficients were extracted from the LUT for each combination of PSD parameters in Table 6 and then put into the bimodal solution algorithm to calculate the inferred PSD parameters. It is worth noting that while the range and resolution of PSD values that were used to extract extinction coefficients from the LUT (Table 6) were reduced, the overall range and resolution of the LUT used to find solutions were consistent with values in Table 5.

Visualization of the performance in bimodal space is challenging because of the number of variables. However, the stability of these results can be elucidated by looking at summary statistics and comparing with the results of the single mode study. These statistics are presented in Table 7.

      Here, the data were binned according to the *inferred* $r_1$ for both the single and bimodal solutions. To build this table we took, for example, all PSD parameters that resulted in an inferred $r_1$ of 100 nm and calculated percentile statistics (i.e., $P_5$, $P_{25}$, $P_{50}$,

$P_{75}$, $P_{95}$). Therefore, when the algorithm infers an $r_1$ of 100 nm for a single mode distribution we are 90% confident that the true (target) value is between 80 and 100 nm (this is in agreement with the results of Sec. 4). Indeed, throughout the "Single Mode Distribution" column of Table 7 the inferred $r_1$ and target $r_1$ values were in good agreement (generally within $\approx 10\%$), demonstrating the stability and accuracy of this method when solving for single mode parameters.

| Parameter | Range/Values | Resolution |
|---|---|---|
| $r_1$ (nm) | 75 – 300 | 25 |
| $r_2$ (nm) | 150 – 800 | 50 |
| $\sigma_1$ | 1.1 – 2.0 | 0.1 |
| $\sigma_2$ | 1.01 – 1.8 | 0.1 |
| $(N_1/N_{total})$ | 1, 0.999, 0.99, 0.975, 0.95, 0.90 | NA |
| Extinction error (%) | 2.5 | NA |

**Table 6.** Range and resolution of target values that were fed into the bimodal solution algorithm for stability testing.

The bimodal solutions are presented in a similar manner but include details about the second mode. For example, it is possible to infer an $r_1$ of 100 nm when the true (target) value of $r_1$ ranges from 100–150 nm and when the true value of $r_2$ ranges from 250–800 nm (the high degree of variability in the target $r_2$ statistics is driven by the varying number density ratios). Overall, the inferred $r_1$ values are in good agreement with the median target values (i.e., the output matches the input) until the inferred $r_1$ exceeded 300 nm. As seen in Table 6 the biggest $r_1$ value we used as input for the solution algorithm was 300 nm, so any inferred $r_1 > 300$ nm should be unexpected. Further, it is interesting to note that in Table 7 the inferred $r_1$ continued to increase even when the target radius remained small, especially as the inferred $r_1$ approached 500 nm. This seemingly unexpected result is due to the presence of larger particles within the second mode dominating the extinction (as discussed in Sec. 3.1.1 and shown in Fig. 13) and thereby opening the solution space to nonsensical values. However, it is important to note that while the irrationality of these results is obvious under controlled simulations, we would have no basis for rejecting these results under real-world conditions. Therefore, we view this as a conclusive demonstration of potentially significant instability in the bimodal solution when the second mode's particles are large and demonstrates the infeasibility of accurately inferring bimodal PSD parameters from SAGE data.

## 5.3 Recommendations for bimodal solution space

The analysis presented in this section demonstrated that bimodal PSD parameters can be inferred from the SAGE extinction spectra and a casual interpretation of Fig. 12 could lead one to conclude that the situation is hopeful. However, this subsection is presented to remove ambiguity on the interpretation of the bimodal evaluation.

First, moving to a bimodal solution space comes at an additional computational cost. Indeed, despite reducing the LUTs' PSD ranges and resolutions, as compared to the single-mode LUTs, the overall runtime increased by more than 2 orders of magnitude. Therefore, this raised an obvious question: are bimodal solutions worth the increased runtime? To address this the reader is encouraged to note the similarity between panels g–l where single-mode PSD parameters were inferred and s–x where bimodal PSD parameters were inferred in Fig. 12. While the bimodal solutions tended to yield a narrower range of solutions (i.e., the shaded regions are narrower in panels s–x), the overall similarity in the median profiles is striking. This indicates that using a bimodal solution space did not significantly improve the overall accuracy of these solutions.

| Single Mode Distribution | | Bimodal Distribution | | | |
| --- | --- | --- | --- | --- | --- |
| Inferred $r_1$ ($\pm 10$ nm) | Target $r_1$ Statistics ($P_5, P_{25}, P_{50}, P_{75}, P_{95}$) | Inferred $r_1$ ($\pm 10$ nm) | Target $r_1$ Statistics ($P_5, P_{25}, P_{50}, P_{75}, P_{95}$) | Inferred $r_2$ ($\pm 10$ nm) | Target $r_2$ Statistics ($P_5, P_{25}, P_{50}, P_{75}, P_{95}$) |
| 80 | (70, 70, 80, 80, 90) | 80 | (100, 100, 100, 137, 150) | 720 | (567, 662, 725, 787, 800) |
| 100 | (80, 90, 90, 100, 100) | 100 | (100, 100, 100, 100, 150) | 650 | (250, 600, 650, 700, 800) |
| 120 | (100, 110, 120, 120, 140) | 120 | (100, 100, 100, 100, 150) | 655 | (350, 600, 650, 712, 800) |
| 140 | (120, 130, 140, 140, 160) | 140 | (100, 150, 150, 150, 150) | 710 | (250, 600, 700, 750, 800) |
| 160 | (150, 150, 160, 160, 180) | 160 | (100, 150, 150, 150, 150) | 660 | (250, 500, 650, 700, 800) |
| 180 | (170, 170, 180, 180, 190) | 180 | (100, 150, 150, 150, 200) | 540 | (300, 387, 500, 600, 750) |
| 200 | (190, 190, 200, 200, 210) | 200 | (100, 150, 200, 200, 250) | 600 | (300, 500, 600, 700, 800) |
| 220 | (210, 210, 220, 220, 230) | 220 | (100, 150, 200, 250, 300*) | 565 | (327, 450, 600, 700, 800) |
| 240 | (230, 230, 240, 240, 250) | 240 | (100, 200, 200, 250, 250) | 630 | (400, 500, 650, 750, 800) |
| 260 | (240, 250, 260, 260, 270) | 260 | (100, 150, 250, 250, 300*) | 600 | (350, 450, 625, 750, 800) |
| 280 | (261, 270, 280, 280, 290) | 280 | (100, 200, 250, 300*, 300*) | 630 | (350, 500, 650, 750, 800) |
| 300 | (280, 290, 300, 300, 310) | 300 | (100, 200, 250, 300*, 300*) | 645 | (400, 550, 675, 750, 800) |
| 320 | (300, 310, 320, 320, 330) | 320 | (100, 100, 250, 300*, 300*) | 510 | (400, 450, 600, 700, 800) |
| 340 | (320, 330, 340, 340, 350) | 340 | (100, 100, 150, 300*, 300*) | 440 | (400, 450, 550, 650, 800) |
| 360 | (340, 350, 360, 360, 370) | 360 | (100, 100, 100, 300*, 300*) | 505 | (400, 450, 500, 700, 750) |
| 380 | (350, 370, 380, 380, 390) | 380 | (100, 100, 100, 225, 300*) | 580 | (450, 450, 550, 700, 800) |
| 400 | (370, 390, 400, 400, 410) | 400 | (100, 100, 100, 100, 100) | 435 | (450, 450, 450, 550, 625) |
| 420 | (390, 410, 420, 420, 430) | 420 | (100, 100, 100, 100, 100) | 430 | (450, 450, 500, 500, 650) |
| 440 | (410, 430, 440, 450, 450) | 440 | (100, 100, 100, 100, 100) | 620 | (500, 500, 500, 600, 720) |
| 460 | (430, 450, 460, 470, 470) | 460 | (100, 100, 100, 100, 100) | 560 | (500, 500, 550, 550, 620) |
| 480 | (450, 470, 480, 490, 490) | 480 | (100, 100, 100, 100, 100) | 450 | (500, 500, 500, 550, 700) |
| 500 | (460, 490, 500, 500, 500) | 500 | (100, 100, 100, 100, 100) | 740 | (550, 550, 550, 600, 650) |

**Table 7.** Mode radius solution percentiles for the single and bimodal distributions. The percentiles ($P_i$) indicate the range of target radii that resulted in the indicated inferred radius. Asterisks indicate values that are limited by the range of target values in the model (see Table 6). The uncertainty in k was fixed at 5%.

Second, we demonstrated that the second mode has a disproportionate impact on the first mode's estimation. While the overall influence of the second mode, and the corresponding statistics in Table 7 can be modulated by varying distribution widths, number densities, and composition (e.g., what if the first mode is 125 nm smoke particles the second mode is 400 nm sulfuric acid particles?), the overall interpretation is clear: while the OPC analysis looks promising the theoretical evaluation in Table 7 demonstrates that there is not enough information within the SAGE extinction spectra to sufficiently constrain the solution space to accurately infer bimodal PSD parameters. Based on these results we conclude that the cost of inferring bimodal PSD parameters is high and the benefit is, at best, modest and recommend against using bimodal PSD solutions based solely on SAGE data.

## 6 Application to SAGE data

Since this manuscript is concerned primarily with the algorithm development and performance we will not make an in-depth scientific investigation of any particular event. Rather, herein we explain the details of the application of this algorithm to the SAGE II and SAGE III/ISS missions and provide a high-level overview of the PSD parameters for these missions. The intention here is to provide the reader with a general overview of the performance of this algorithm and the variability of the PSD parameters over the lifetime of these instruments. Having established the methodology, more targeted scientific analyses will be the subject of subsequent publications.

### 6.1 SAGE II

The PSD algorithm was applied to data collected under the SAGE II mission using only the 520:1021 nm extinction ratio (i.e., condition #0 in Table 4) to infer single-mode lognormal distribution parameters. The 384 nm channel was excluded following the guidance of Damadeo et al. (2013) and the 448 nm channel was not included as it is relatively close to the 525 nm channel and did not significantly improve the performance. As shown above (i.e., Figs. 7 and 9) the accuracy of the inferred PSD values is inversely proportional to the measurement uncertainty. Therefore, extinction coefficients with uncertainty >20% were excluded from the analysis. Further, a simple cloud-filter was applied, following Thomason and Vernier (2013), by excluding all extinction ratios $\leq 1.4$ (we note that cloud filtering did not impact the aggregate statistics shown below).

The reported $k_{1020}$, median inferred PSD parameters, and the median inferred microphysical properties during the SAGE II time period are presented in Figs. 15 and 16 (northern and southern hemispheres, respectively). Notable volcanic eruptions, as defined in Table 8 are indicated with labels above panel (a) in both figures as well as vertical dashed lines within each panel. Here, it is observed that the record was dominated by two major events: the 1982 eruption of El Chichón and the 1991 eruption of Pinatubo. Both eruptions resulted in bimodal aerosol distributions (Knollenberg and Huffman, 1983; Oberbeck et al., 1983; Deshler et al., 1992, 1993, 2003), which means that the inferred PSD parameters will be heavily weighted toward the coarser mode as discussed in Sec. 4.4. However, the general observation within these figures is that these eruptions led to overall larger particles (including $r_e$), enhanced SAD, enhanced VD, and enhanced N.

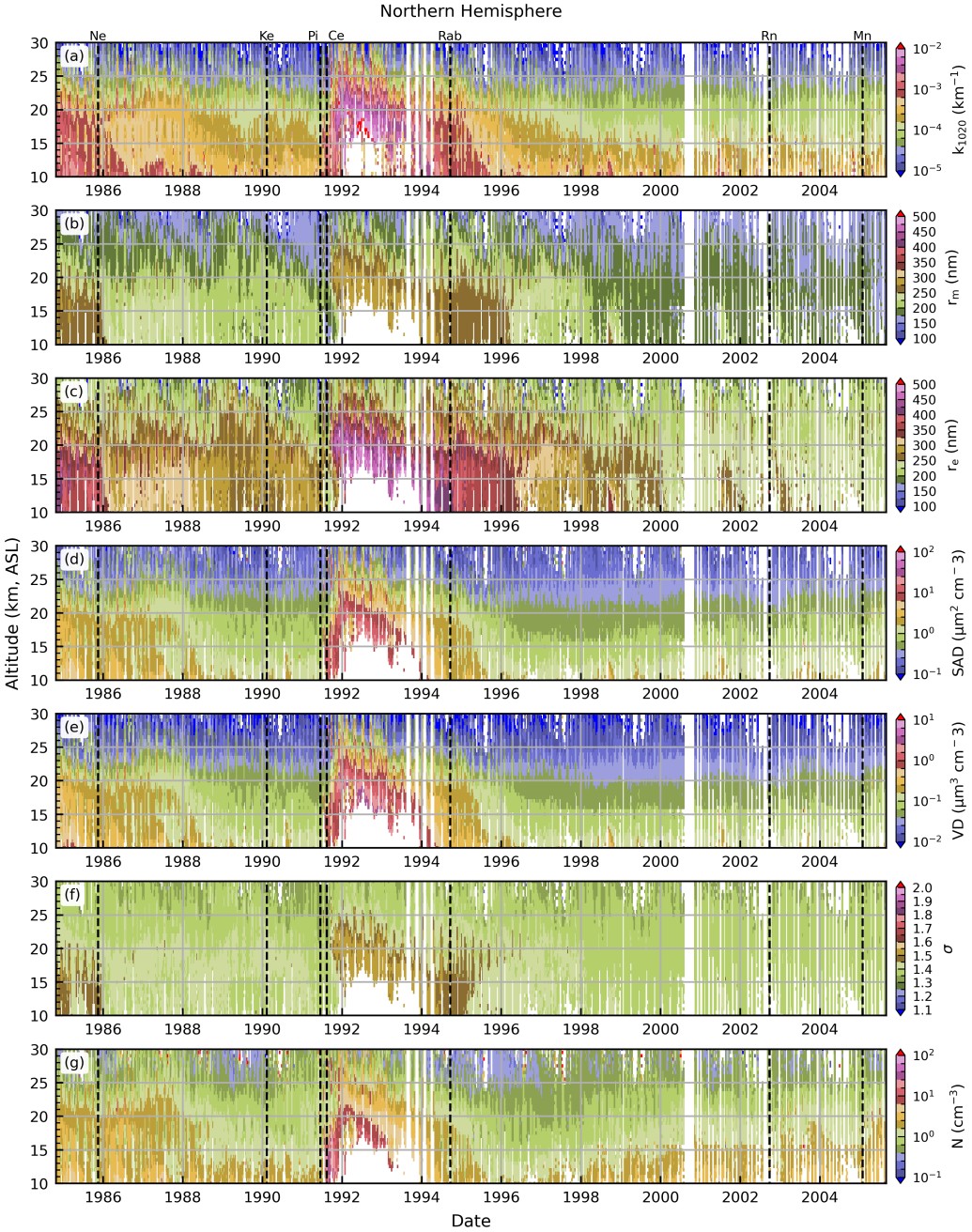

**Figure 15.** Extinction coefficient ($k_{1020}$), median inferred PSD parameters, and the median inferred microphysical properties for SAGE II data collected in the northern hemisphere. The data were filtered to remove extinction coefficients that had an uncertainty in excess of 20% or were indicative of cloud contamination (i.e., $k_{520}/k_{1020} \leq 1.4$).

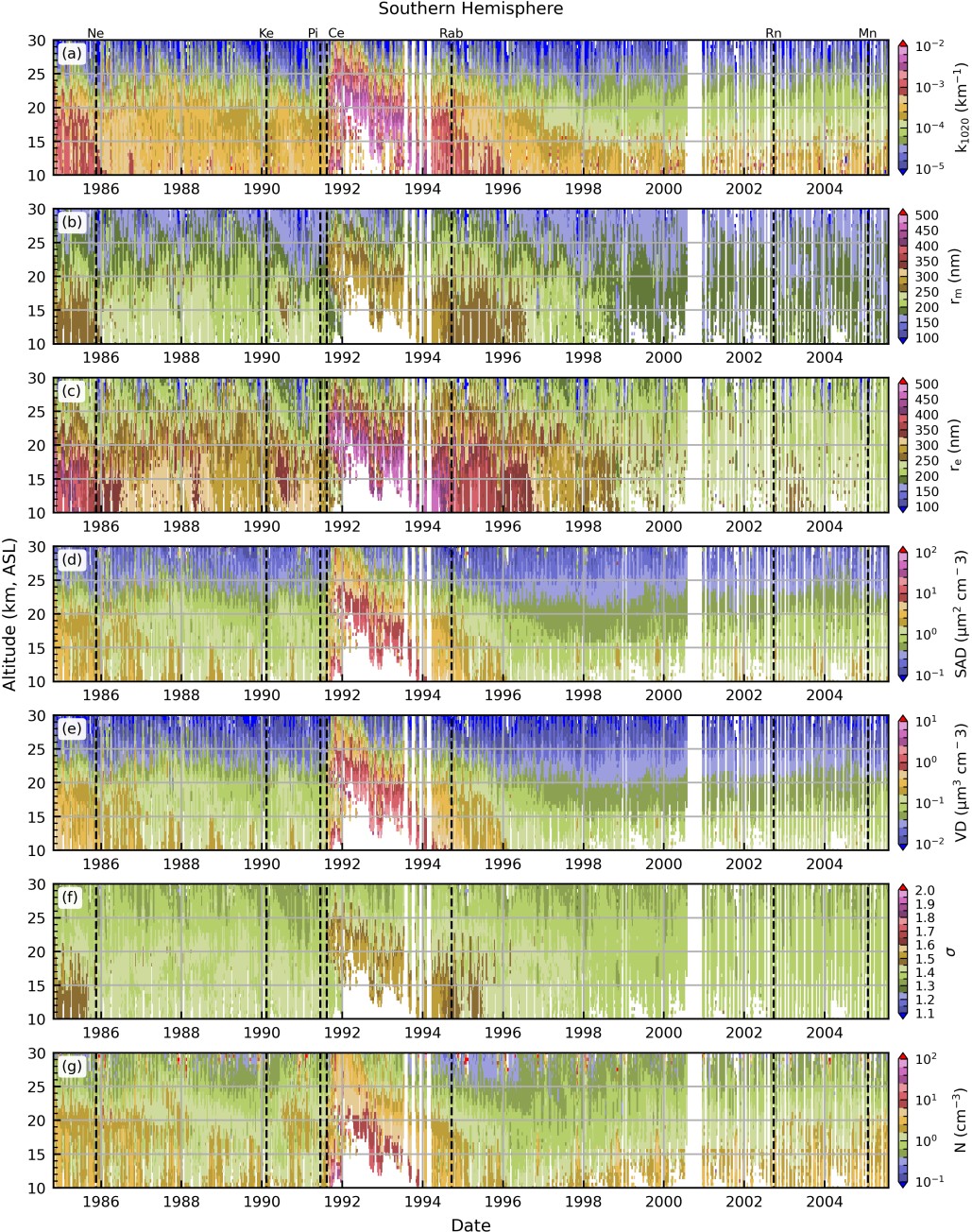

**Figure 16.** Same as Fig. 15, but for the southern hemisphere.

| Event Name | Date | Latitude |
|---|---|---|
| Nevado del Ruiz (Ne) | November 1985 | 5°S |
| Kelut (Ke) | February 1990 | 9°S |
| Pinatubo (Pi) | June 1991 | 15°N |
| Cerro Hudson (Ce) | August 1991 | 46°S |
| Rabaul (Rab) | September 1994 | 4°S |
| Ruang (Rn) | September 2002 | 2°S |
| Manam (Mn) | January 2005 | 4°S |
| Canadian pyroCb (Cw) | August 2017 | 52°N |
| Ambae (Am) | July 2018 | 15°S |
| Raikoke (Ra) | June 2019 | 48°N |
| Ulawun (Ul) | June 2019 | 5°S |
| La Soufriere (LS) | April 2021 | 13°N |
| Australian pyroCb (Aw) | January 2020 | 27-35°S |
| Hunga Tonga (HT) | January 2022 | 20°S |

**Table 8.** Notable volcanic and pyroCb events during the SAGE II and SAGE III/ISS records. Table includes labels used to identify events in Figs. 15, 16, 18, and 19.

The SAGE II v7.0 product files contained SAD and $r_e$ estimates as described in Thomason et al. (2008) and Damadeo

et al. (2013). Thomason et al. (2008) provided a detailed discussion on the uncertainty in the inferred SAD values, which is generally within ±30% in the main aerosol layer but may have an overall range in excess of 200% under light aerosol loads. The agreement between the current algorithm's SAD and $r_e$ products, as compared to the v7.0 products, is shown in Figure 17. Data in Fig 17 were filtered using the same criteria as in Figs. 15 & 16. Here, panels (a) and (d) present normalized histograms of the abundance of the v7.0 SAD and $r_e$ products, respectively. Panels (c) and (f) present the relative abundance of the percent

difference between the current algorithm's SAD and $r_e$ estimates and the v7.0 values, respectively. Finally, panels (b) and (e) present scatter plots, color coded by $k_{1020}$, of the percent difference between the 2 algorithms vs the v7.0 SAD and $r_e$, respectively.

Two general observations are made within Fig. 17: First, as $k_{1020}$ increased so too did SAD and $r_e$; Second, as $k_{1020}$ increased the agreement between the 2 algorithms improved. Further, under enhanced aerosol load (e.g., $k_{1020}$>1E-4 km$^{-1}$) the majority

of the SAD and $r_e$ estimates were within the ±30% uncertainties stated in Thomason et al. (2008). The histograms in Fig. 17 show the relative distribution of these physical parameters as well as the percent difference. It is observed that more than half of the SAD estimates were greater than 1 $\mu$m$^2$ cm$^{-3}$ (panel a) and the $r_e$ had a broader range (panel d) with 4 modes at 180, 230, 260, and 300 nm.

Overall, the PSD estimates from SAGE II were comparable to values observed within the OPC record (see e.g., Deshler et al.,

2003) and the SAD and $r_e$ estimates agree with the v7.0 products within the stated uncertainty under enhanced conditions. While

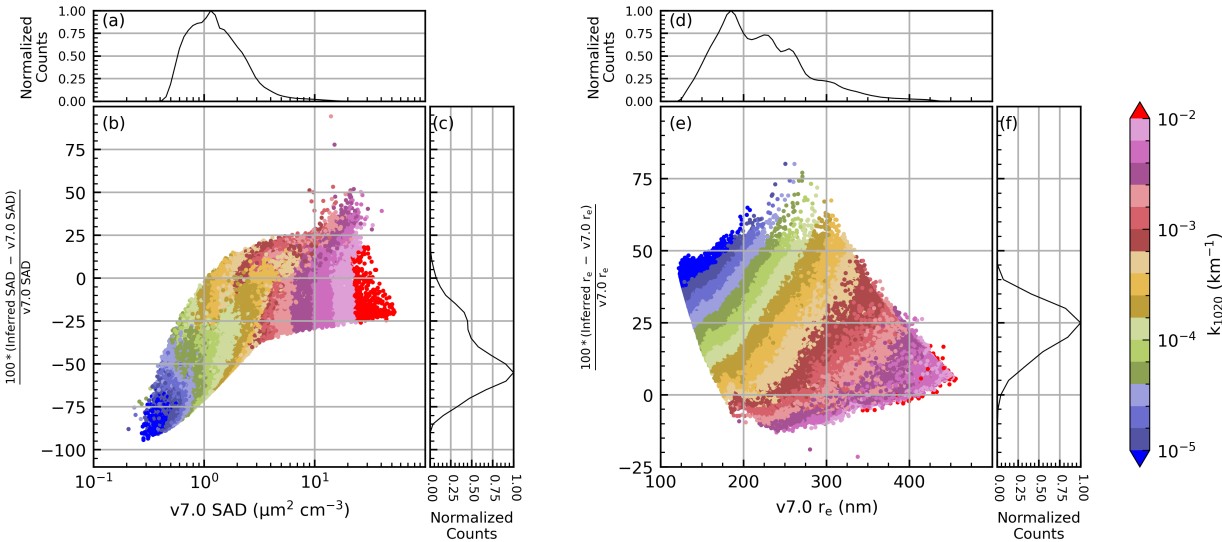

**Figure 17.** Comparison between the SAD and $r_e$ estimates from the current algorithm and those in the v7.0 SAGE II product. Data within these panels were filtered using the same criteria used in Figs. 15 & 16.

this does not provide definitive validation of this new algorithm this does demonstrate consistency between this new method and an established product that is currently in use within the community and provides confidence in validity of this technique.

## 6.2 SAGE III/ISS

As discussed in Sec. 4, a combination of condition definitions were used for identifying the PSD parameters in the SAGE III/ISS record. First, solutions were searched using condition #5 (i.e., all channels except those in the Chappuis band). There are several reasons why this condition would fail to deliver valid solutions, all of which relate to the input extinction coefficients. For example, if the reported extinction for any channel is negative or a channel is saturated then the solution algorithm cannot perform. In the case that a solution is not found using condition #5 the algorithm then excludes the 384 nm channel (condition #6) and makes another attempt to find solutions. The exclusion of the 384 nm channel is, of course, based on the assumption that the failure of condition #5 was due to the 384 nm channel being either negative or saturated. Finally, in the case that condition #6 fails to find a solution the algorithm switches over to condition #15 (i.e., the third best performing condition per Sec. 4, data not shown) and makes a final attempt to find the PSD solutions. If no solutions were found in condition #15 then no PSD parameters were provided for that particular spectrum. After finding solutions the data were filtered to remove the influence of cloud contamination using the method of Kovilakam et al. (2023). We note that the changes between conditions did not result in step-function-like changes within the PSD profiles.

The results are shown in Figs. 18 and 19 for the northern and southern hemispheres, respectively, for the entire SAGE III/ISS record. The northern hemisphere was dominated by the 2017 Canadian wildfire and the 2019 eruption of Raikoke as

well as smaller eruptions (e.g., the 2021 eruption of La Soufriere) and transport from larger events that occurred in the southern hemisphere (e.g., the 2022 eruption of Hunga Tonga). The southern hemisphere was dominated by the 2020 Australian wildfires as well as the 2022 eruption of Hunga Tonga (more discussion below). Indeed, the impact of the Canadian and Australian wildfires persisted for ≈1 and ≈2 years, respectively. This resulted in large particle and narrow distribution width estimations throughout these time periods. However, as discussed above, these results must be interpreted with caution given the uncertain composition, phase, and morphology of smoke particles.

Both hemispheres showed a consistent pattern of smaller particles formed immediately after the smaller eruptions (i.e., Ambae, Ulawun, La Soufriere) while the larger eruptions produced larger particles. This small particle formation was directly correlated with increased number density (panel g) and inversely correlated with the distribution width (panel f). However, the Raikoke and Hunga Tonga eruptions, as well as the major wildfire events, consistently produced larger particles with smaller distribution widths. While these events did not yield number densities comparable to the smaller events the number densities remained elevated as compared to background conditions. These results are in agreement with previous studies (Thomason et al., 2021; Wrana et al., 2023).

### 6.2.1 Hunga Tonga case study

We now present a brief demonstration of the algorithm's products in the aftermath of the 2022 eruption of Hunga Tonga. A more thorough scientific evaluation will be the subject of a subsequent publication (in progress). Here, we take the opportunity to finish the presentation of the statistics that are output from this method and do a cursory comparison to balloon-borne optical particle counter observations.

Monthly zonal median plots of the inferred effective radii are presented in Fig. 20 wherein it is observed that the main plume contained particles with $r_e > 400$ nm, in March 2022, and this plume was centered near 10°S and 24 km. The plume was transported predominantly southward though some of the particles were caught in the natural northward circulation. While all particle sizes were transported to lower altitudes as a product of the Brewer-Dobson circulation (especially outside tropical latitudes) the larger particles showed a disproportionately rapid descent, as compared to the smaller particles. This is most readily seen in comparing panels a–d with panels e–h of Fig. 20 where, regardless of latitude, we observed a partitioning between the largest particles (>400 nm) and smaller particles (<300 nm) over the ensuing year. Here, the plume of largest particles was centered at ≈20 km near the equator and descended to lower altitudes toward the higher latitudes. This is in contrast to the persistence of smaller particles at higher altitudes, well past 25 km. Finally, we note the apparent growth of particles with time, predominantly at lower altitudes, where $r_e$ approached 500 nm in March/June 2023.

The PSD estimates from an individual profile collected on 15-February 2022 are shown in Fig. 21. Before discussing the content of this figure it is important to recall the method we used in inferring the PSD parameters (i.e., extinction coefficient LUTs were used to identify *all* theoretical PSD parameters that yielded extinction ratios within the propagated uncertainty of the SAGE III/ISS extinction ratios). This enabled us to provide a statistical representation of the PSD solution space on a point-by-point basis, part of which is the overall spread in the solution space. This statistical representation is presented in Fig. 21 with the spread in the solution space represented by the shaded regions.

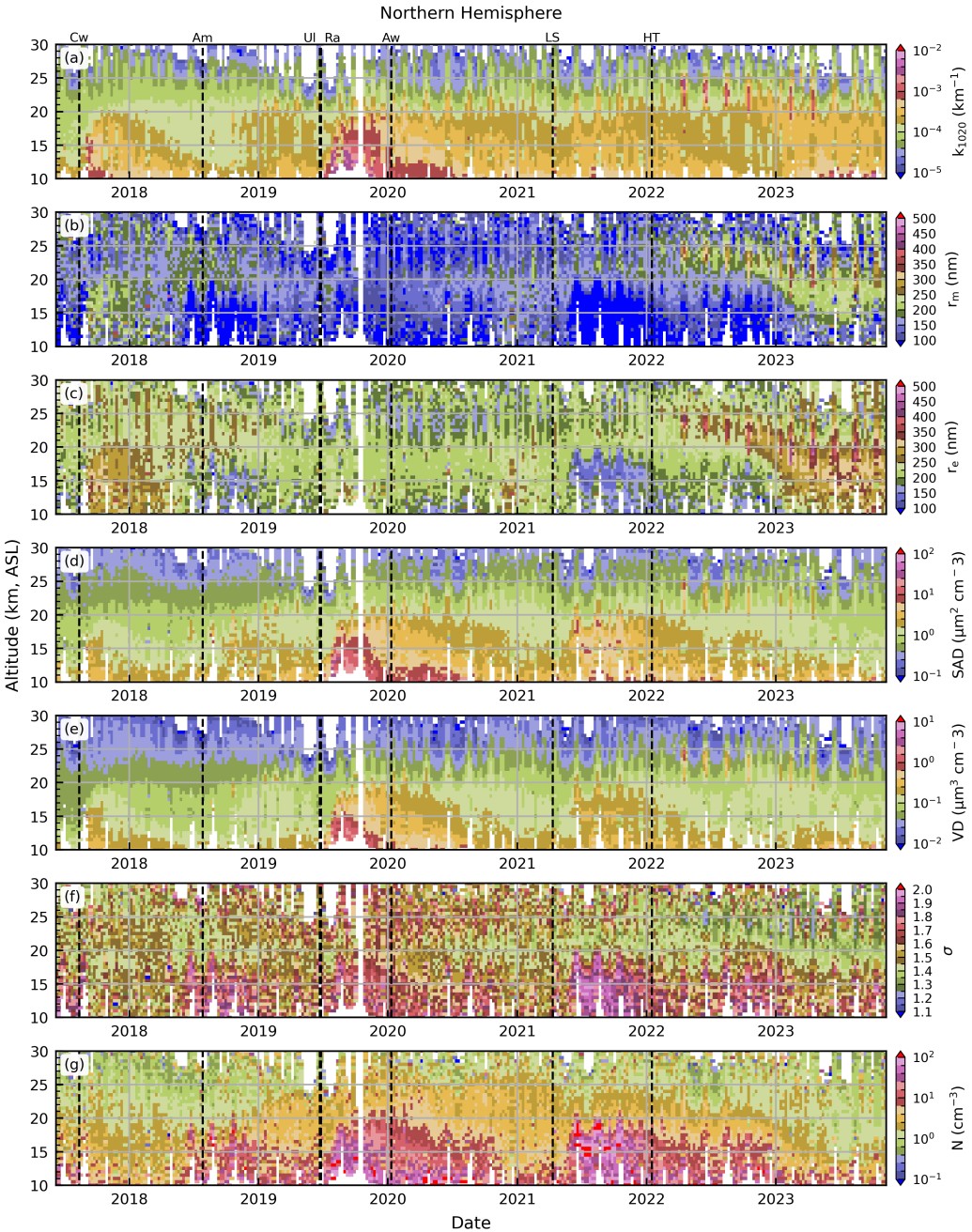

**Figure 18.** Extinction coefficient (1020 nm, panel a) and size distribution parameters (panels b-e) inferred over the entire SAGE III/ISS record for the northern hemisphere.

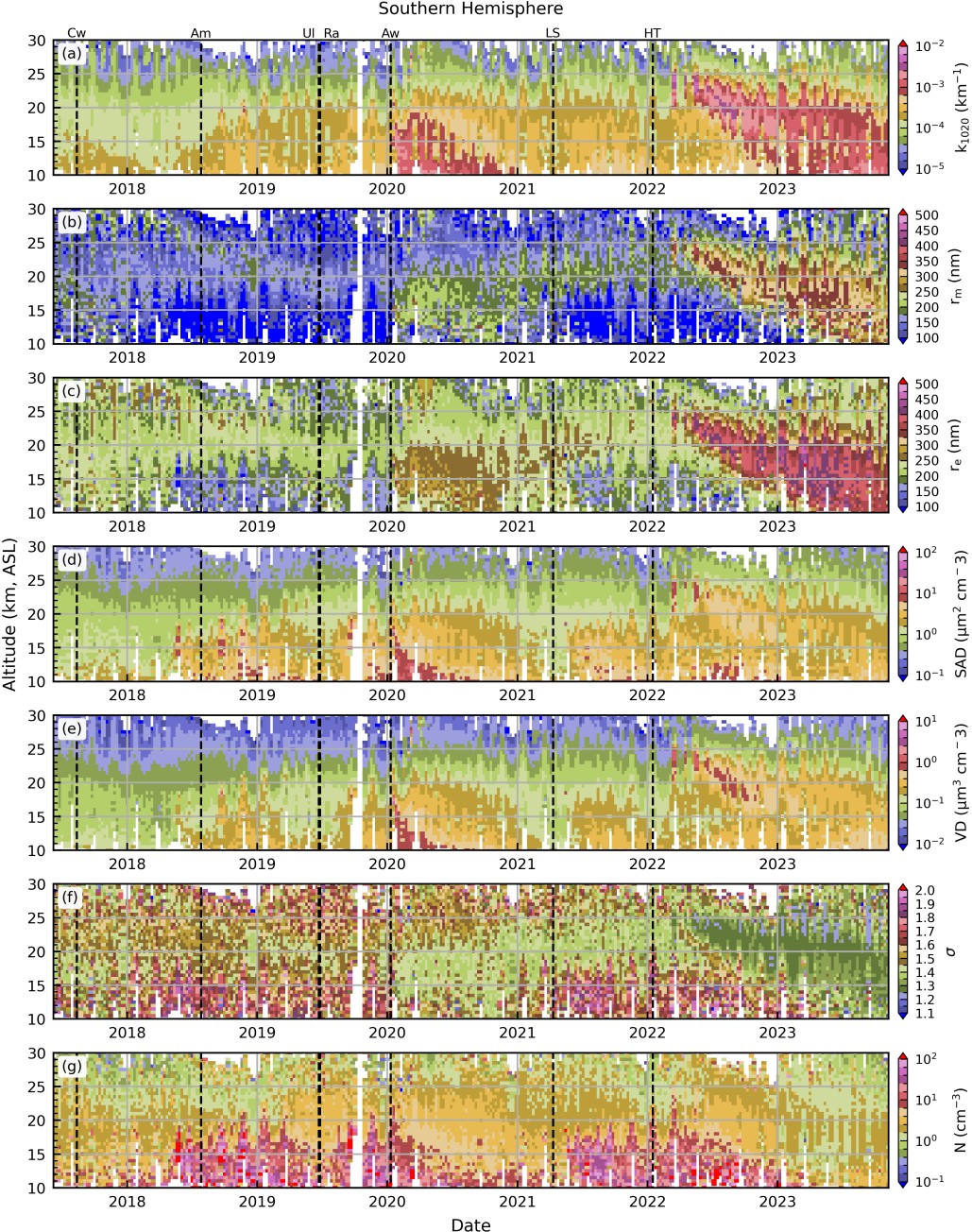

**Figure 19.** Same as Fig. 18, but for the southern hemisphere.

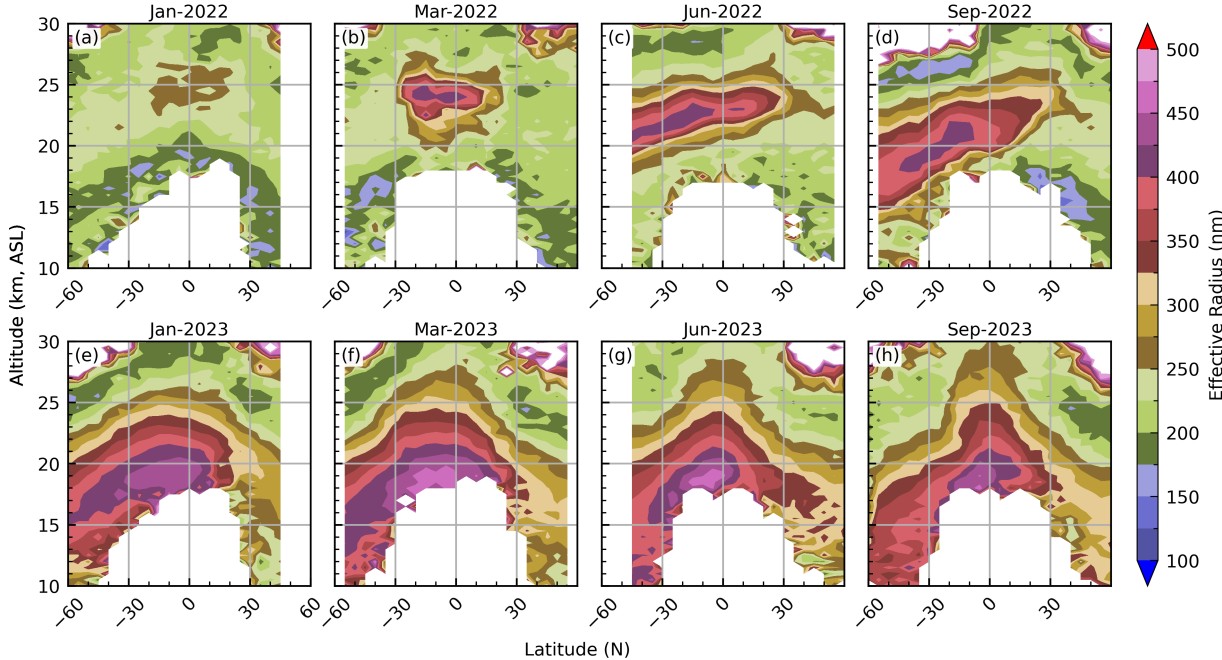

**Figure 20.** Monthly zonal median effective radii in the months following the Hunga Tonga eruption.

Panel (a) of Fig. 21 contains the extinction coefficient profiles for 3 wavelengths and panel (g) contains the percent errors of these coefficients. Panels (b) through (f) contain profiles of the inferred parameters with the shaded region indicating the median $\pm$MAD[*]. Panels (h) through (l) contain the 90% confidence interval (CI) relative to the median (vertical dashed lines are guides to the eye and represent the $\pm$10% and $\pm$20% levels).

Three distinct peaks were observed in the extinction profile at 18 km, 21.5 km, and 24 km, though the solution algorithm failed to find any PSD parameters that yielded extinction ratios comparable to the SAGE III/ISS values below 19 km. Further, the spread in solution space, due to increased extinction uncertainty (panel g) near 20 km resulted in less certain PSD estimates. However, the reported uncertainty in the extinction coefficients at 24 km was smaller and resulted in a narrower range of PSD parameter solutions. The estimated mode radius at 24 km was 345 nm $\pm$20 nm and a distribution width ($\sigma$) of 1.3 $\pm$0.05. We note that the inferred $r_e$ value at 24 km (410 nm $\pm$5 nm) is in good agreement with the estimate of 460 nm from data collected by a Portable Optical Particle Spectrometer (Asher et al., 2023) as well as other recent independent estimates of $r_e$ (Khaykin et al., 2022; Legras et al., 2022; Duchamp et al., 2023). In summary, the variability in the extinction profiles correlates well with variability in the PSD, SAD, VD, and $r_e$ profiles.

The CI statistics are a key feature of this methodology (panels h through l) and provide guidance on the level of confidence of these estimates. For example, looking at the mode radius data at 24 km we conclude that 90% of the theoretical PSD parameters that matched the SAGE III/ISS extinction ratios (within the bounds of the reported uncertainty) had a mode radius and distribution width that were within 10% of the inferred median value. A similar statement can be made for the SAD, VD,

and $r_e$ estimates as well. The utility of this method is that this CI estimation opens an opportunity for implementing not only the PSD and microphysical products in chemistry and climate models, but this opens the opportunity for improved uncertainty estimates for the input parameters that drive these models as well as improved uncertainty estimates for the products that come from these models.

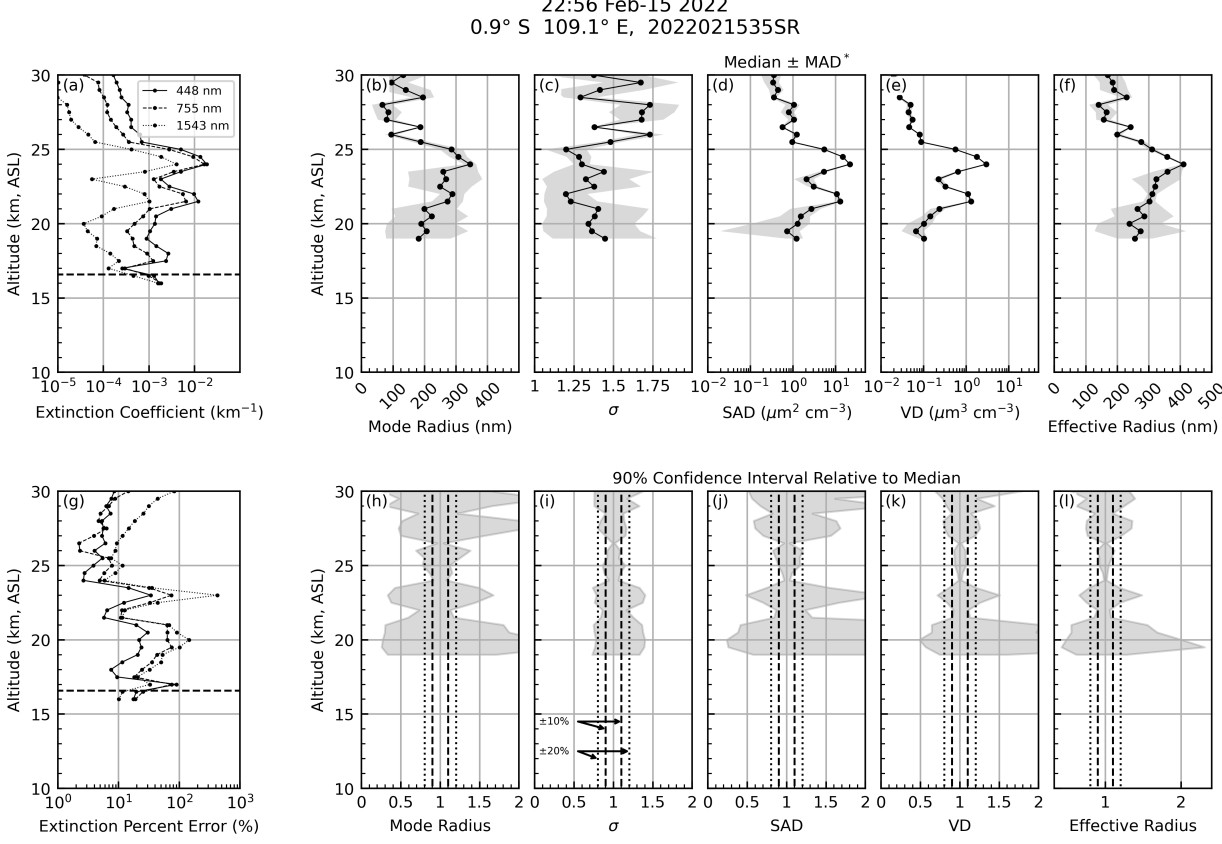

**Figure 21.** Extinction profiles (a), extinction percent error profiles (g), PSD profiles (b-f), and 90% confidence interval profiles (h-l). The horizontal dashed line in panels (a, g) represents the tropopause altitude.

## 6.3 Putting the SAGE II results into perspective

As discussed above, the PSD algorithm was applied to the SAGE II record using the condition #0 criteria (i.e., a single extinction ratio). While the overall qualitative behavior of those results, as shown in Figs. 15 and 16 were reasonable, the question remains as to the quantitative reliability of these estimates. To evaluate this we found the PSD solutions for the SAGE III/ISS data using the condition #0 criteria (hereafter referred to as the "cond0" version) and compared these results with the

values calculated using the hybrid condition #5/6/15 of Sec. 6.2 (hereafter referred to as the "hybrid" version). The results of this evaluation are presented in Fig. 22 for the southern hemisphere.

Here, it is observed that the 2 versions are generally with $\approx$30% of each other for the higher-moment parameters. The distribution width ($\sigma$) had, by far, the best agreement where the entire record was within 2%. However, the $r_m$ estimate had mixed results outside the main aerosol layer. Indeed, the cond0 $r_m$ estimates were >50% higher than the hybrid estimates for much of that record. Finally, the N estimates for cond0 were notably lower (i.e., higher percent difference) outside the main aerosol layer. We note that these large percent differences were driven by variability of small numbers. While this results in apparent wild fluctuations on a relative scale, the absolute variability is small.

Bulk statistics, that represent both the northern and southern hemispheres are presented in Table 9. These statistics demonstrate a systematic over-estimation in the $r_m$ and $r_e$ estimates, which the SAD, VD, and $\sigma$ differences had medians closer to zero and inter-quartile ranges that were within $\approx \pm 25$% of the median. The overall interpretation of Table 9 and Fig. 22 is that the cond0 estimates are in good agreement with the hybrid results within the main aerosol layer and certainly under elevated aerosol load (e.g., post Hunga Tonga). However, the agreement is notably worse outside elevated aerosol conditions. These results are consistent with the findings of Thomason et al. (2008) regarding their discussion of SAD and $r_e$ estimates (both of which relied on 2 wavelengths). While these results are encouraging for estimating post-eruption PSD parameters, we were unable to determine a distinct cutoff in $k_{1020}$ above which the cond0 estimates were consistently in good agreement with the hybrid products. Finally, while the cond0 products may be useful for observing general trends within the SAGE II record, users must be aware of this product's limitations.

| Parameter | NH Statistics $(P_{10}, P_{25}, P_{50}, P_{75}, P_{90})$ | SH Statistics $(P_{10}, P_{25}, P_{50}, P_{75}, P_{90})$ |
|---|---|---|
| $r_m$ | (-214, -95, -28, 3, 21) | (-191, -74, -15, 12, 24) |
| $r_e$ | (-70, -36, -13, 0, 8) | (-65, -31, -8, 3, 9) |
| SAD | (-19, 1, 20, 43, 60) | (-21, -7, 12, 39, 58) |
| VD | (-14, -2, 8, 21, 33) | (-14, -5, 5, 19, 32) |
| $\sigma$ | (-13, -3, 4, 14, 22) | (-15, -7, 2, 12, 21) |
| N | (-68, -4, 47, 80, 93) | (-79, -29, 32, 75, 92) |

**Table 9.** Percentile statistics, divided by hemisphere, for the percent difference between the hybrid and cond0 product. The percent difference was calculated via 100 * (hybrid - cond0) / hybrid.

## 7 Conclusions

We presented a methodology for inferring particle size distribution (PSD) parameters from SAGE extinction spectra. The novelty of this methodology is in the statistical representation of the PSD solution space, which provides the community with the

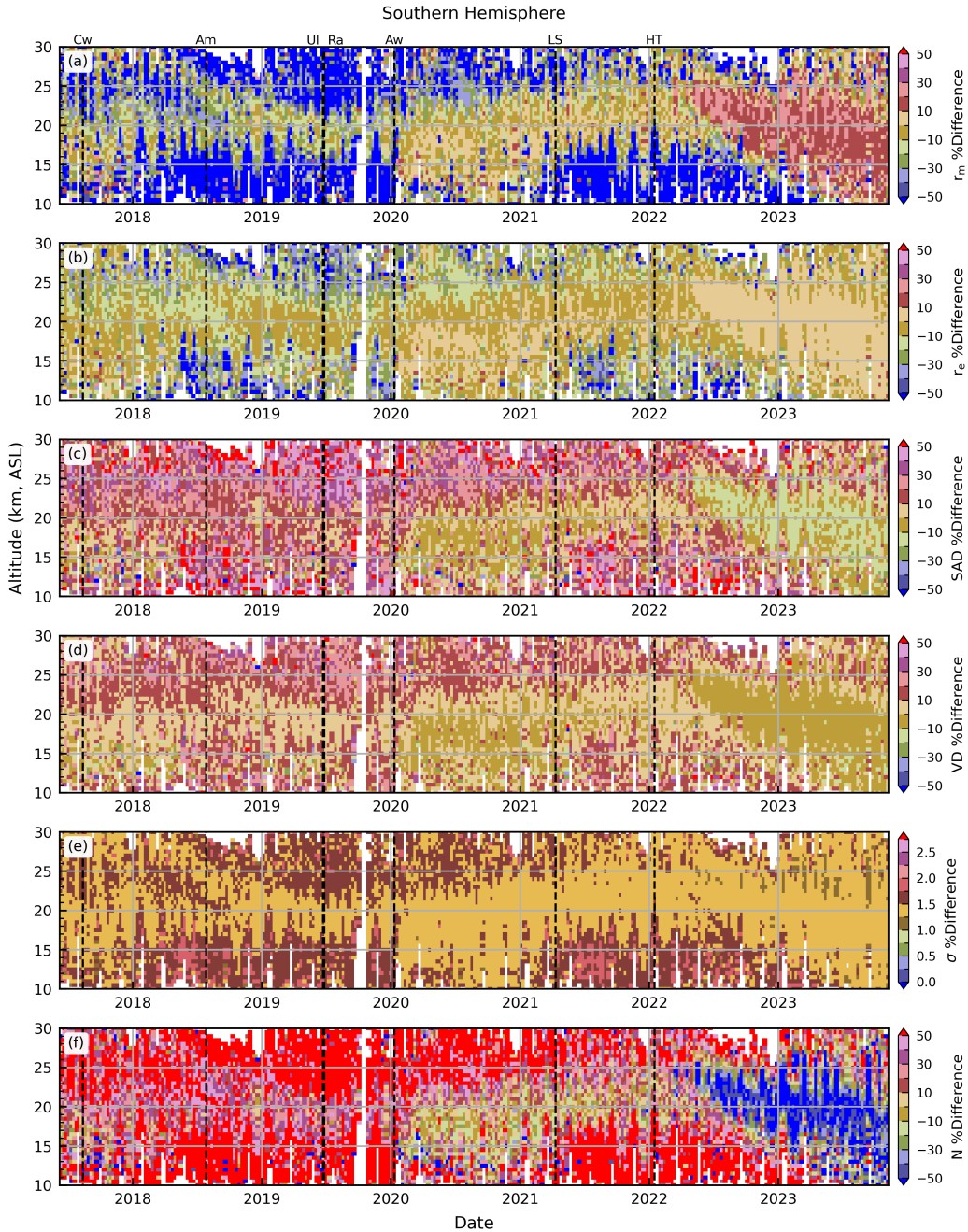

**Figure 22.** Percent difference between PSD products inferred using the hybrid condition #5/6/15 and those inferred using only condition #0. The percent difference were calculated by taking the 100 * (hybrid - cond0) / hybrid.

potential range of PSD and microphysical values as well as the corresponding confidence intervals. While the PSD solutions are valuable for climate and chemistry modelling, the statistical information provides an additional level of information previously not available. Indeed, this additional information allows end users to propagate these uncertainties through their respective calculations and thereby improve the error assessment of their end products.

The accuracy of this method was evaluated in terms of both single mode and bimodal lognormal distributions as a function of extinction coefficient uncertainty. We demonstrated that the inferred single-mode PSD parameters were within $\approx \pm 25\%$ when the extinction error was small (Figs. 7 & 9). Further, we evaluated the impact that incorrect composition assumptions had on the single-mode PSD estimates (Figs. 10 & 11). A key result of this study is that while PSD parameters can be inferred using extinction spectra from instruments such as SAGE these parameters cannot be represented by a single number (e.g., mean or median). Rather, the extinction spectra, within the bounds of measurement uncertainty, can be reproduced by myriad PSD combinations and this variability must be reported as done here.

The PSD and microphysical property products were compared to the University of Wyoming's OPC record. Overall the two records were in good agreement.

We studied the feasibility of obtaining bimodal solutions from the SAGE III/ISS spectra alone (Sec. 4.4) and used the University of Wyoming's OPC record as a case study to evaluate the accuracy of these results. While the bimodal solution algorithm returned PSD estimates that were in generally good agreement with the OPC record this came at the cost of significantly increased computation time and with the caveat that the solution space was heavily biased toward coarse mode particles (i.e., the algorithm tried to minimize $r_1$ and maximize $r_2$). In effect, the algorithm is forcing all of the finer-mode particles into a regime that SAGE is insensitive to (e.g., $\lesssim 150$ nm per Fig. 2) so that the fine mode, in effect, could be ignored. Indeed, the coarse-mode particles dominated even when the fine-mode particles outnumbered them by more than 100:1. We made a detailed discussion regarding the physical reasoning behind this bias. Based on this analysis we concluded that the bimodal solution space is too unstable to provide consistently reliable PSD estimates without incorporating an additional dataset.

PSD values for both the SAGE II and SAGE III/ISS data were obtained for the entire data record and the variability of these parameters was demonstrated. Herein we showed that the SAGE II record was dominated by events that yielded large particles throughout most of its record while the SAGE III/ISS record is composed of a mixture of events (volcanic and pyroCb) that resulted in both smaller and larger particles as discussed by Wrana et al. (2023). Our surface area density (SAD) and effective radius ($r_e$) estimates for the SAGE II data were compared to the standard SAD and $r_e$ products within the SAGE II v7.0 product. While the v7.0 products and our products were within the specified uncertainties (i.e., $\pm 30\%$ under enhanced aerosol load and $\pm 200\%$ under background conditions per Thomason et al. (2008)) the overall agreement was better under enhanced conditions.

The SAGE II PSD parameters were inferred using only a single extinction ratio (condition #0), which calls into question the quality of these estimates due to the limited information content of 2 wavelengths. The performance of condition #0 was evaluated by creating PSD estimates from the SAGE III/ISS record using both condition #0 (cond0) and the original hybrid of conditions (using conditions #5, 6, 15). Here, it was determined that the cond0 estimates were regularly larger than the hybrid estimates, which led to a skew in the overall distribution.

The SAGE III/ISS PSD parameters, the higher-moment parameters, and the uncertainty estimates represent new SAGE III/ISS level 2 products and are available to the community for download and use (NASA, 2024). Please see Data Availability section for details.

*Data availability.*  The SAGE data used within this study are available on NASA's Atmospheric Science Data Center (ASDC; https://eosweb.larc.nasa.gov/). The SAGE III/ISS PSD parameters are available at NASA's ASDC (NASA, 2024) as stand-alone files until they are merged into an upcoming major release of L2 product. At that point the PSD data will be within the main SAGE III/ISS record and the stand-alone files will no longer be updated.

*Author contributions.*  TNK developed the methodology, wrote the solution algorithm and wrote the manuscript. LT and MK aided in interpretation of the results and application to the Hunga Tonga eruption. SJM provided guidance on the algorithm's architecture and optimization. All authors participated in scientific and algorithmic discussions and all authors reviewed the manuscript prior to submission.

*Competing interests.*  The authors declare that they have no competing interests.

*Acknowledgements.*  This work was funded as part of the SAGE III/ISS Science Team. SAGE III/ISS is a NASA Langley managed mission funded by the NASA Science Mission Directorate within the Earth Systematic Mission Program. Enabling partners are the NASA Human Exploration and Operations Mission Directorate, International Space Station Program and the European Space Agency. ADNET Systems Inc. personnel are supported through the NASA LARC RSES contract 80LARC17C003.

The authors would like to express their gratitude to Mr. Felix Wana for reading the pre-submitted version of this manuscript and providing insightful comments that improved this manuscript prior to submission. The authors would also like to thank Dr. Chris Boone and the anonymous reviewers for reviewing this manuscript and providing actionable feedback regarding the quality and content of the submitted version of this paper. Finally, the authors thank Dr. Sandip Dhomse for leading the review of this paper as the handling editor.

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
