# Peer review of "Characterization of Stratospheric Particle Size Distribution Uncertainties using SAGE II and SAGE III/ISS Extinction Spectra"

_Atmospheric Measurement Techniques, 2023_

## Referee Comment (RC1)

This is a novel approach to deriving stratospheric sulfate aerosol information from the SAGE series of instruments.  Overall, the approach seems logical and appropriate.  One of the major benefits of employing a statistical approach is that error estimates automatically 'fall out' of the analysis.  One of the issues I perceive in the results is a possible bias to larger particle size, presumably from larger particles 'leaking' into the solution set, as I will discuss in more detail in specific comments provided below.

Another (probably unavoidable) issue with the approach is that it treats all aerosols as sulfates, even if they are smoke, PSCs, volcanic ash, ice…  It might require a sophisticated user to interpret (or know when to ignore) parameters derived from events other than volcanoes that inject aerosols into the stratosphere, or when there are clouds (e.g., PSCs or cirrus).  However, it is also true that the majority of aerosols in the stratosphere (outside of PSCs in the polar vortex) are likely sulfate, particularly when you consider background aerosols.  The authors provide a warning regarding smoke, but it might be worth extending that warning to include any aerosols other than sulfates.

I will make the comment that sulfate aerosol particle size distribution information derived from SAGE could represent a valuable data set.  Instruments that measure in limb-scattering mode (such as OSIRIS and OMPS) need to assume size distribution parameters in their analysis.  The field seems to have settled on using parameters appropriate for "fine mode" sulfate aerosols (as determined by particle counter measurements) for analysis, based on the argument that fine mode aerosols typically significantly outnumber "coarse mode" aerosols in the atmosphere.  However, larger particles provide dramatically higher contribution to the scattering signal, as this paper shows, and it is not appropriate to simply ignore larger particles when analyzing scattering-based measurements.  The field (whether it realizes it or not) is in desperate need of more representative sulfate aerosol size distribution parameters if we want to determine the impact of sulfate aerosols on climate more accurately.  The SAGE series of instruments represents a likely source for such information.

Specific comments:

**line 126: it was observed that if σ =1.5 (Fig. 4, panel b), $r_m$=75 nm (i.e., background conditions)**

These numbers are typical values ascribed to background sulfate aerosols, but as I said above, they are not representative of the PSD you would need to reproduce measurements involving scattering using a monomodal size distribution.  They are representative of the fine mode in a bimodal size distribution (they are basically the same values you use for the fine mode in Figure 13b when considering the impact of having a bimodal distribution).  You are free to use any numbers you want in your calculations, of course, but explicitly referring to these values as "background conditions" serves to perpetuate the notion that they are representative of the values you would determine when analyzing your background measurements with a monomodal distribution.  They are not.

Looking at Figure 15 (Northern Hemisphere for SAGE II), one would estimate a typical value for $r_m$ under background conditions to be somewhere in the vicinity of 200 nm.  Looking at Figure 18 (Northern Hemisphere for SAGE III/ISS), one might estimate a value closer to 150 nm.

This paper makes the case that enhanced scattering from larger particles means one cannot ignore the coarse mode despite the lower number density relative to the fine mode.  Suggesting that $r_m$ = 75

nm, $\sigma$ = 1.5 is representative of background aerosols undermines that argument because those numbers were chosen to be consistent with fine mode parameters from a bimodal distribution observed by particle counters. Using those numbers argues that the coarse mode can be ignored in measurements involving scattering.

In the end, the goal is not to accurately model the physical nature of atmospheric aerosols, but rather to model the optical characteristics of those aerosols, which is the important consideration for their impact on climate. If the real distribution is bimodal (or some higher order multimodal) and a monomodal analysis is employed, the derived parameters represent effective values for $r_m$ and $\sigma$. The values of these effective parameters will be skewed mightily by the extreme sensitivity of scattering effectiveness as a function of particle size. Larger particles will contribute to the values of the effective parameters in a manner that is wildly out of proportion to their relative number density. That is a limitation of the measurement system that must be acknowledged if we hope to properly interpret the implications of the measurements.

Dealing with effective parameters will muddle the interpretation of the chemical impact of the aerosols (the weighting of the effective $r_m$ to larger values will likely imply a larger aerosol volume than exists in reality). It will also impact the inferred greenhouse effect from absorption in the infrared (absorption and scattering have different sensitivities to particle size). However, those issues are worries for another day. A positive first step would be moving away from PSD parameters that are a good representation of the bulk physical characteristics for the majority of sulfate aerosols in the atmosphere (i.e., $r_m$ = 75 nm and $\sigma$ = 1.5, corresponding to only the fine mode aerosols) to effective PSD parameters that provide a more accurate representation of the optical response of all (e.g., both fine mode and coarse mode if the distribution is bimodal) sulfate aerosols in the atmosphere (i.e., $r_m$ larger and $\sigma$ TBD but probably a bit larger to encompass both modes in the bimodal distribution).

**Line 215: For example, if the reported errors of the extinction coefficients were all within 5% and the inferred mode radius was 100 nm then, per Fig. 7 (c), we know that on median the inferred mode radius is ≈5% too high, and that 90% of the time the inferred value is within ±15% of the target value.**

Looking at Figure 7c, I would suggest we can infer more than this. There is a strong tendency for the analysis to generate a larger value for $r_m$. For $r_m$ below ~200 nm, this is evident even in the average values, with a clear high bias introduced into the inferred value. For $r_m$ above ~250 nm, the effect manifests as a tall upper 'whisker' stretching the fringe of the distribution in the solution set to include larger particles than were present in the pseudo-SAGE data used as the target.

Conversely, the values for inferred distribution width $\sigma$ in Figure 7f are generally biased low (other than for $\sigma$ <= 1.2). This is likely not a coincidence. There is an inverse relationship between $r_m$ and $\sigma$. If you increase one of the parameters, you can compensate for that in the calculation by decreasing the value of the other parameter. It looks like there is some aspect of the sensitivity study that trades a smaller distribution width for a larger mode radius. For smaller $r_m$ (< 250 nm), it doesn't appear to be a straight swap, though, because in Figure 7o the inferred values for effective radius $r_e$ are also biased high. I would call this problematic. Even though the bars in Figure 7o cross 1.0 and the offsets could therefore be deemed to be statistically insignificant, there is a clear bias introduced. These are synthetic

data where you know the truth. I assume that the indicated errors are uncertainty levels assumed in the $\sum$ matrix rather than random errors added to the set of extracted data.

My interpretation would be that the analysis approach tends to allow a number of solutions at larger $r_m$ to leak into the solution set. If it happens in the synthetic data, I expect it would affect the analysis of real measurements as well (i.e., create a bias toward larger $r_m$). Is there any way to adjust the analysis to reduce that bias?

**Line 268: Since the $H_2SO_4$ content of atmospheric aerosols is, ultimately, unknown we note that this situations adds an unknown element to the analysis**

The $H_2SO_4$ content in the droplet will be governed by thermodynamic equilibrium. The $H_2O$ vapor pressure for the droplet should equal the ambient partial pressure of $H_2O$ in the atmosphere (Steele and Hamill 1981, https://doi.org/10.1016/0021-8502(81)90054-9). If one knows the ambient temperature and $H_2O$ concentration, an estimate of the $H_2SO_4$ content can be calculated (Bernath et al 2023, https://doi.org/10.1016/j.jqsrt.2023.108520). If there is higher $H_2O$ pressure in the ambient air, the droplet should take up $H_2O$ to find equilibrium (thereby decreasing wt% $H_2SO_4$). Conversely, if the $H_2O$ pressure in ambient air is lower, there would be evaporation from the droplet to achieve equilibrium.

**Line 280: the particles were composed of 75% $H_2SO_4$ with $r_m$=75 nm, and σ=1.5 (i.e., background conditions)**

Another reinforcement of fine mode parameters supposedly being representative of background aerosols. A caveat of some sort would be appropriate, unless you truly believe the numbers are representative (despite what your later results suggest).

**Section 4.3: observations of stratospheric smoke**

I don't disagree with what was done in this section, but it may underestimate the impact of smoke on the PSD parameters. Unlike the liquid droplet sulfate aerosols, smoke particles are not constrained to be spherical, which could significantly affect its scattering characteristics.

Also, aged smoke particles apparently turn glassy, which would presumably change its optical constants, possibly outside the range of combined BC and BrC employed in the study, although for all I know turning glassy might push the optical constants closer to sulfate aerosol values, which would reduce the systematic errors.

**Figure 12: calculating values for N**

When you are working with extinction ratios, it is not clear how number density (N) is being derived. Information on number density cancels when you calculate the ratios. Even if you were working with straight extinction values (rather than ratios), you would only be able to determine the column density of

the particles along the line of sight (the average number density times the path length through the sulfate aerosols), not the number density itself.

A discussion of adding number density to LUTs occurs in Section 5. I assume that is involved. The ratio of number density for the two modes in a bimodal distribution will affect the shape of the spectrum, so I can see how that ratio could be determined in the bimodal analysis. Nothing in the paper indicates how it is being done in a monomodal analysis where you are working with extinction ratios.

**Line 339: However, the shorter wavelengths show an interesting behavior in that their extinction increased rapidly followed by a flattening and slight decrease in extinction, which is subsequently followed by another rapid increase.**

This behavior is actually associated with resonances in the Mie scattering for certain particle sizes. Note that the curve for 384 nm in Figure 13 has two peaks, one near where $r_2$ is roughly equal to the wavelength (around 384 nm), and one where $r_2$ is roughly twice the wavelength (around 768 nm). Similarly, the curve for 520 nm has a peak where $r_2$ is near the wavelength (520 nm). If you were to extend the plots to higher wavelength, you would presumably see a second peak somewhere around 1040 nm.

**Line 441: what if the second mode has a different composition than the first?**

Droplets with different sizes will have slightly different $H_2O$ vapor pressures because it will be impacted by the degree of curvature of the surface, but I would expect the composition of two droplets under the same conditions (temperature and ambient $H_2O$ concentration) to have relatively comparable compositions.

**Line 479: under enhanced aerosol load (e.g., $k_{1020}$>1E-4 km$^{-1}$) the majority of the SAD and $r_e$ estimates were within the ±30% uncertainties stated in Thomason et al. (2008)**

We can also note from Figure 17 that the values of $r_e$ from this study are systematically higher than the v7.0 results, which is perhaps another indication that this approach yields a high bias, particularly at smaller mode radius, as was (to me) suggested by the sensitivity study results in Figure 7.

**Line 503: The southern hemisphere was influence by more events including transport from the 2017 Canadian wildfire**

It is surprising that you would choose not to mention Australian Black Summer (or Australian New Year, as it is also called) wildfires from 2019/2020, labeled Aw in Table 7, the most dramatic Southern Hemispheric event in recent years other than the Tonga eruption. It would certainly rate mention before long-range transport from the 2017 Canadian wildfires (more typically termed the Pacific Northwest pyroCb event). The impact of the 'Aw' event is clearly evident in $r_m$ and $\sigma$ in Figure 18. See, for example, the light green swath in the plot for $\sigma$ that extends from the beginning of 2020 into early 2022, at which

point σ in the stratosphere turns dark green following the Tonga eruption.  It looks like enhanced smoke aerosols from Aw might have persisted for a couple of years.  This would be an opportunity to warn readers that because these are smoke aerosols, not sulfate, the values of the parameters will have systematic errors.

**SAGE II versus SAGE III/ISS**

No comparison is made between the SAGE II results (Figures 15 and 16) and the SAGE III/ISS results (Figures 18 and 19).  Later (in the Conclusion), the statement is made that the SAGE II recorded was dominated by large particles, but I would suggest it is a stretch to claim that background aerosols were unusually large for the entire duration of the SAGE II data record.  There is presumably an aspect of differences in analysis approach for the two instruments contributing here.

In general, the SAGE III/ISS results show more structure, which is likely expected from the additional spectral information available.  As alluded to in the Conclusion, SAGE II appears to yield larger values of $r_m$ for background aerosol conditions.  It is worth asking whether that is a byproduct of using only one extinction ratio in the analysis (condition #0).  It might be instructive to analyze the SAGE III/ISS data with the same approach as was used for SAGE II (using condition #0) and look at the differences from the nominal results (using the hybrid conditions 5/6/15).  That would give a sense of potential systematic errors in the SAGE II results arising from the limited spectral data employed in the analysis.

**Line 523: Here, the plume of largest particles were centered at ≈20 km near the equator and descended to lower altitudes toward the higher latitudes.**

The difference between the peak altitude near the equator versus that at higher latitudes is not a direct indication of aerosol sedimentation.  As was mentioned earlier in the paragraph, Brewer Dobson circulation will naturally transport air to lower altitude as it moves poleward, even in the absence of sedimentation.  To get a sense of the descent, you should observe the location of the > 300 nm plume at a particular latitude (e.g., 30 °S or 45 °S) and note how it changes over time.

Minor comments and typos:

**line 45: As an occultation measurement, the SAGE instrument peers through hundreds, sometimes thousands, of kilometers of atmosphere**

It depends on how you define the extent of the atmosphere, I suppose.  For measurement purposes, I would define the overall height of the atmosphere according to the range over which atmospheric extinction contributes significantly to the signal.  With that consideration, I would say you are looking through hundreds of kilometers, not thousands.

**line 95: "distribution width"**

This term was never explicitly defined. Looking at Equation 2, some people might interpret distribution width as $\sigma$, while others might interpret it as $\ln(\sigma)$. The latter choice might be the more logical interpretation. $\sigma$ would be literally interpreted as the width of the distribution in $\ln(r)$ space. $\sigma$ is defined in the paper as the geometric standard deviation, and the text shifts into referring to it as distribution width without comment.

**Table 1: Range of $H_2SO_4$ composition**

Ideally, when determining a statistical projection of the solution set, your 'basis set' would span the full set of expected conditions. At low temperatures and/or high $H_2O$ concentrations, wt% $H_2SO_4$ can be 50% or lower (Bernath et al 2023, https://doi.org/10.1016/j.jqsrt.2023.108520). These would fall outside the chosen basis set.

**line 110: Logically, 1 can infer**

Traditionally, it would be written as "one can infer"

**line 205: to determine how well to 2 matched**

Typo. "…how well the two matched"?

**Figure 7: Plots are being provided for VD, a quantity that has yet to be mentioned in the text and doesn't get mentioned for some time.**

**Line 254: LUTS**

LUTs

**Line 268: Since the $H_2SO_4$ content of atmospheric aerosols is, ultimately, unknown we note that this situations adds an unknown element to the analysis**

situations -> situation

**Line 335: $r_1$=75 nm, $\sigma_1$=1.45, $r_2$=310 nm, $\sigma_1$=1.05**

$\sigma_2$=1.05

**Caption of Figure 15: $k_{1020}/k_{1020} \leq 1.4$**

$k_{520}/k_{1020} \leq 1.4$

**Caption of Table 7: Table includes labels used to identify events in Figs. 18 and 19.**

And Figures 15 and 16

**Line 503: The southern hemisphere was influence by…**

influence -> influenced.

**Line 521: due to differing deposition rates**

deposition -> sedimentation?

**Line 523: Here, the plume of largest particles were centered at ≈20 km**

were -> was (plume is singular).

**Line 567: Overall the 2 records were in good agreement.**

2 -> two

---

## Author Comment (AC1)

We would like to thank this reviewer for reviewing this manuscript and providing valuable feedback. Our responses are provided below (blue) to the reviewer's comments (black).

Page 13, line 233: insert "" before conditions 5 and 6. Corrected

Figure 8: Why does the fraction decrease when errors > 20% are removed? I would assume the opposite case: The more accurate data in a data set, the more valid results. What exactly is the denominator of the presented fractions?

The denominator is the total number of extinction spectra in the SAGE record (for a given latitude/altitude). Therefore, filtering data inevitably reduces the throughput. However, the intent of this figure is to, in part, determine what the overall throughput is for these 3 conditions and determine how much the throughput is reduced by only using data with the lowest uncertainty. The caption of this figure was updated to reflect how the fractions were calculated.

Figure 8: I can clearly see a distribution of the fraction. However, I can hardly distinguish between fractions smaller than 0.7 (line 226) and larger than this limit. Maybe the authors could use a stronger color gradient?

The color scale was modified to enhance contrast.

Figure 9: Maybe the authors could show this Figure before Fig. 8?

We agree that grouping the figures together like this may make sense. However, the figures are listed in the order in which they appear in the text, per the Copernicus guidelines. We made no changes in regard to this comment.

Page 14, line 245: What are "challenging aspects of real-world aerosol compositions and PSD parameters"? Please specify.

The text was updated to provide the reader of some typical examples.

Page 16, line 264: Please give definition of SAD and VD.

The abbreviations SAD and VD are defined within the paper. A table was added to the paper (Table 3 of the new version) to provide a mathematical definition of these terms.

Page 19, line 296: Please specify used UWY OPC data set (e.g. time frame, location, number of profiles, influence of volcanic aerosols, . . . )

The requested information was added.

Sec. 4.4.2 – 5.3: I am not convinced that one can compare the PSD parameters of the first mode of a bimodal distribution with the parameters of a monomodal PSD and draw conclusions about the quality of the retrieval. E.g., page 24, line 389: "though $r_2$ was underestimated by $\approx 90\%$."

How can something be underestimated if the reference does not even exist? PSD parameters of a bimodal curve and PSD parameters of a monomodal curve are completely different things. In this case, comparisons can only be made on the basis of integrated parameters (SAD, VD, re).

An excellent point. We think the issue the reviewer raised is in regard to Section 5.1 only. All discussion that precedes Sec. 5.1 is critical to the discussion as this relates to the reliability of a single-mode estimation when the atmospheric aerosol is really bimodal. This is critical.

On the other hand, Sec. 5.1 begins the discussion of bimodal solutions within the context of the OPC case studies. We agree that the wording here is ambiguous. The intent of the text that the reviewer quoted (i.e., "though r2 was underestimated by $\approx 90\%$.") was to inform that reader that the inferred PSD parameters for the second mode were substantially different from the OPC's first-mode parameters (e.g., the algorithm isn't saying that $r_1$ and $r_2$ have the same value) ... and the solution algorithm's second mode is not dominating the solution space (e.g., the algorithm isn't trying to minimize the influence of the first mode and forcefully fit all of the OPC spectra into the second mode).

The text was updated to make this clear.

Figure 12: How is N retrieved?

Another reviewer raised a similar concern. Subsection 3.2.3 was added to discuss this. In short, the PSD parameters within the solution space were used to calculate the 1 $\mu$m extinction coefficient with N set to 1 cm$^{-3}$ and the observed 1 $\mu$m extinction coefficient was divided by the calculated values, which indicates the difference in scale (i.e., N). Equation 2 was also updated to explicitly show that N plays a role.

Page 21, line 335: typo: r1=, sigma1=, r2=, sigma1= -¿ sigma2
Corrected

Figure caption 13: Please specify OPC record.

We believe the information that we added, in response to another of the reviewer's concerns, addresses this question. However, we updated the caption to point the reader to the text for details.

Page 30, line 478: "1. ...,2. ..." -¿ First, ..., second, ...
Changed

Page 30, line 484: "PSD estimates are reasonable" -¿ What is this statement based on?
The text was updated.

Page 31, line 495-499: Do the authors see a "jump" in the retrieved data when they change the conditions within the retrieval?

No. The text was updated to inform the reader of this nuance.

---

## Author Comment (AC2)

We would like to thank this reviewer for reviewing this manuscript and providing valuable feedback. Our responses are provided below (blue) to the reviewer's comments (black).

There are no questions regarding the results because the conclusions are predicated on assumptions established during computation.

This is not a correct interpretation. While we do not interrogate every possible assumption regarding the inference of aerosol size distributions using SAGE III data, we do, and in fact it is the main point of the paper, examine in depth the impact of key assumptions in this process. So in this regard, we are not certain exactly what this reviewer is objecting to.

I believe the title should be revised, adding "Characterization of Stratospheric..."

Stratospheric was added to the title.

The affiliation list for the third author, 2,1, seems odd. Switching it to 1,2, if it is associated with both institutes, appears more appropriate.

The affiliation list was updated.

Is this approach not a revisiting or improvement of Wrana et al.'s Stratospheric Aerosol Size Distribution Retrieval from SAGEIII/ISS?

To some degree yes, though these types of estimates have been done for decades (as cited in the paper). The Wrana et al. paper was discussed within this manuscript and we included discussion on improvements we sought to make (including an extensive analysis of the impact of the assumptions).

Section 2.1 seems like a less significant subsection. Yes, the computational strategy remains a fundamental aspect of data loading, processing, and modeling. In open science research, limitations in computational resources should not serve as excuses for achieving suboptimal results. Nevertheless, it is ultimately up to the authors to decide what they choose to present.

It is unclear to us what portion of our work the reviewer is referring to as "suboptimal" but we hopefully address this misunderstanding here. First, the purpose of this statement, within the context of this manuscript, it to tell the reader that reproducing the single-mode analysis does *not* require expensive hardware such as the A100. However, the bimodal analysis does require the bigger A100. There were no trade offs made here and the hardware selection in no way limited the accuracy of the results.

Line 44: The challenge of performing "traditional" ????

The text was updated to include examples of traditional validation work.

"65%, 70%, 75%, and 80% sulfuric acid by weight." At what conditions are all these propositions of sulfuric acids present in the stratospheric aerosols? 75% is conventional acceptance for

background aerosols. May be going up to 90% just after significantly impactful volcanic eruptions makes more sense, especially when SGAE signals are not penetrating below tropopause or so.

This is possible. However, the purpose of this section is to evaluate the impact of incorrect composition assumptions. To that end, we believe the section fulfilled its purpose. The reviewer makes a good point about the variability of sulfuric acid content especially in the wake of large volcanic injections. As another reviewer pointed out, the sulfuric acid content can be estimated based on collocated temperature and water vapor measurements. Currently, the SAGE mission produces a water vapor product, but the temperature product is still in development. When the temperature product becomes available we can possibly modify our algorithm to estimate sulfuric acid weight percent. However, such an endeavor will be a separate study.

Do the black carbon and brown carbon are uniformly present in the stratospheric aerosols? If so, is the Mie theory still suitable for this situation? Yes, the possibility of the existence of absorbing aerosols may be considered, but it is not guaranteed solely based on sporadic occultation extinction.

As discussed in the paper, stratospheric smoke is a huge unknown. The composition, etc. is largely undetermined. Again we note that the intent of this section is to determine the impact of an incorrect composition assumption. To answer the reviewer's question: yes, Mie theory is an appropriate model for smoke particles.

Also, "65%, 70%, 75%, and 80% sulfuric acid by weight." So, what constitutes the remaining fraction of water vapor, black or brown carbon, or unknowns?

Yes, the rest of the particle is composed of water. The text was updated to reflect this.

Also in line 250: "H2SO4 is typically assumed to be 75%."" Correct. We see no required change in regard to this comment.

It seems appealing to use black or brown carbon, but it cannot ignore the complex chemistry in the stratosphere at higher temperatures and shorter wavelengths of radiation, as well as the ozone at its maximum. So, we need to be mindful while making such arguments with limited resources.

We apologize, but it is unclear to us what the reviewer means by this comment. The SAGE retrieval algorithm accounts for diverse components in the stratosphere (including ozone) and the aerosol extinction coefficients were derived in light of these components.

Line 148: $(0.3\pm15\%, 1.2\pm10\%)$ It should be explained: the basis of 0.3 and 1.2 to be selected as representative and any reasoning for selecting these two ratios. Is there any information provided by these ratios (just an arbitrary random number or something else)?

Correct, these ratios were not taken from the SAGE record. Rather, they are referred to in the text as being "nominal" values to inform the reader that these are reasonable ratios but not representative of a specific measurement. As described in the text, this section is an illustration of the general solution process. Since the text already describes these values as "nominal" we see no necessary change.

Shorter wavelengths (448 nm) are susceptible to the other molecules. So, the ratio obtained by using them is reliable and consistent?

This depends on where the observations occur within the atmosphere. As discussed in the text, the shorter wavelengths (including 448 nm) attenuate higher in the atmosphere than the longer wavelengths. Conversely, the signal in the longer-wavelength channels drops below the noise floor at higher altitudes. This was one of the factors in our decision to limit the altitude ranges and in our selection of the hybrid "condition" selection (specifically section 4.1). Since the paper already addresses this issue we do not see any changes to be made.

In Figure 5 (a), The ratios on the x-axis place the larger wavelength's extinction in the numerator, while on the y-axis, the smaller wavelength occupies the numerator position. If there's no specific reason for this arrangement, it might be more consistent to use the same approach for both ratios and incorporate their respective values unless there is an otherwise to do otherwise.

The reviewer is correct that there is no scientific basis for putting one wavelength above the other. However, this presentation is consistent with that presented in Wrana et al. 2021 (indeed, this specific section addresses the Wrana et al. method, including limitations) so we keep this the same for consistency.

Same also in line 240 (#5, #6, and #15) from table 3 applied further.

It is unclear what the suggested change is. The manuscript was not changed in regard to this comment.

Lines 459 and 499: The cloud is filtered. The aerosol product is already cloud (opaque) filtered. Is it not? (SAGE III/ISS documents suggest that.) Don't we miss out on the fresh, larger particles after the eruption during cloud filtering? Did your results show a significant difference between cloud-filtered and non-cloud-filtered?

This is not correct. The SAGE data are not already cloud filtered and it is unclear what documentation the reviewer is referring to. If there is an ambiguity or incorrect statement in the retrieval documentation then we would kindly ask the reader to bring this to the attention of someone on the SAGE team.

The filtering algorithm is discussed in a publication by Mahesh Kovilakam (cited in the paper). Indeed, distinguishing between thin clouds and thick aerosol can be challenging. Undoubtedly removing clouds will change the PSD estimation on individual points within individual profiles (i.e., cloud filtering removes data). However, we did not notice any substantive change in the aggregate products after cloud filtering. This is now explicitly stated in the paper.

Figure 15 caption: last line: "(i.e.,k1020/k1020≥1.4)" Now corrected

---

## Author Comment (AC3)

We would like to thank Dr. Boone for his close reading of the manuscript and providing valuable feedback. Our responses are provided below (blue) to the reviewer's comments (black). Because some of Dr. Boone's comments are quite long we abbreviate them with the ellipsis (i.e., ...) when subsequent text is further elaboration on a single, overarching, point.

These numbers are typical values ascribed to background sulfate aerosols, but as I said above, they are not representative of the PSD you would need to reproduce measurement involving scattering using a monomodal size distribution...

This is correct and an excellent point. In general the stratospheric aerosol community have operated under the notion that particle size distributions (PSDs) are monomodal. While OPC observations clearly indicate that the distributions are bimodal (sometimes tri-modal), inferences of PSD parameters from remote sensing observations are almost always forced into the single-mode regime due to lack of information (we do note Dr. Boone's recent work on solving for bimodal PSDs using SAGE III/ISS and ACE-FTS spectra). This has led to many in the field referring to "background" PSDs of $r_m$=75 nm and $\sigma$=1.5...which in truth refers only to the first mode. Further, Dr. Boone is correct that a major thrust of this paper is that these larger modes cannot be ignored except under certain conditions (e.g., wavelength and number density dependent). We agree with Dr. Boone that a paradigm shift needs to be made.

Before proceeding we would like to make the point that the text quoted by Dr. Boone (i.e., line 126 of the original manuscript) was in reference to determining the lookup table (LUT) resolution required to sufficiently mitigate the impact that the resolution has on the inferred PSD parameters. Indeed, under the current LUT design this results in LUT resolutions that are too high (i.e., the impact, as shown in Fig. 4, is likely much lower). However, Dr. Boone's point is still valid that a single-mode distribution of $r_m$ = 75 nm and $\sigma$=1.5 is not the best representation of background conditions. The text in the manuscript was updated to reflect this.

...My interpretation would be that the analysis approach tends to allow a number of solutions at larger $r_m$ to leak into the solution set. If it happens in the synthetic data, I expect it would affect the analysis of real measurements as well (i.e., create a bias toward larger $r_m$). Is there any way to adjust the analysis to reduce that bias?

Dr. Boone's interpretation of the interplay between $r_m$ and $\sigma$ is correct regarding the inverse relationship and how an overestimation of one may offset the other. Unfortunately the information content within the SAGE data itself is too limited to further reduce these biases. We do note that, per the discussion in the original manuscript, the bulk of these biases are withing $\pm \approx 15\%$ when the measurement uncertainty is within 5% (not uncommon for SAGE extinction spectra). This bias decreases as particle size increases. We do not view this as a bad situation.

As discussed in the original manuscript we evaluated several extinction ratio combinations as well as variations on the LUTs' PSD ranges. While the various LUT variations (see Table 2 of the original paper) resulted in PSD estimates that were, generally speaking, in good agreement, the LUT parameters presented in the paper are what yielded the minimum bias.

The $H_2SO_4$ content in the droplet will be governed by thermodynamic equilibrium. The $H_2O$ vapor pressure for the droplet should equal the ambient partial pressure of $H_2O$ in the atmosphere (Steele and Hamill 1981, https://doi.org/10.1016/0021-8502(81)90054-9). If one knows the ambient temperature and $H_2O$ concentration, an estimate of the $H_2SO_4$ content can be calculated (Bernath et al 2023, https://doi.org/10.1016/j.jqsrt.2023.108520). If there is higher $H_2O$ pressure in the ambient air, the droplet should take up $H_2O$ to find equilibrium (thereby decreasing wt% $H_2SO_4$). Conversely, if the $H_2O$ pressure in ambient air is lower, there would be evaporation from the droplet to achieve equilibrium.

This is a fair and valuable point. The intent of this section was to demonstrate the impact of assuming an incorrect $H_2SO_4$ content. Therefore, we see no need to make a functional change to the algorithm. However, we incorporated a similar caveat in the paper and now include the suggested references.

Another reinforcement of fine mode parameters supposedly being representative of background aerosols. A caveat of some sort would be appropriate, unless you truly believe the numbers are representative (despite what your later results suggest).

Addressed above for a previous comment. Here, the reference to "background conditions" was removed. Since the purpose of this section is to evaluate the impact of an incorrectly assumed composition there is no further modification to the text.

I don't disagree with what was done in this section, but it may underestimate the impact of smoke on the PSD parameters. Unlike the liquid droplet sulfate aerosols, smoke particles are not constrained to be spherical, which could significantly affect its scattering characteristics. Also, aged smoke particles apparently turn glassy, which would presumably change its optical constants, possibly outside the range of combined BC and BrC employed in the study, although for all I know turning glassy might push the optical constants closer to sulfate aerosol values, which would reduce the systematic errors.

We agree that the composition, shape, etc. of stratospheric smoke is largely unknown. We acknowledge this uncertainty in the paper. Because of this uncertainty we did not make any changes to the paper. Indeed, the final sentence of this section ("Therefore, we must provide a cautionary note when using data that may be contaminated by the presence of smoke.") was provided to raise this awareness.

When you are working with extinction ratios, it is not clear how number density (N) is being derived...

Another reviewer raised a similar concern. Subsection 3.2.3 was added to discuss this. In short, the PSD parameters within the solution space were used to calculate the 1 $\mu$m extinction coefficient with N set to 1 cm$^{-3}$ and the observed 1 $\mu$m extinction coefficient was divided by the calculated values, which indicates the difference in scale (i.e., N). Equation 2 was also updated to explicitly show that N plays a role.

This behavior is actually associated with resonances in the Mie scattering for certain particle sizes. Note that the curve for 384 nm in Figure 13 has two peaks, one near where $r_2$ is roughly equal to the wavelength (around 384 nm), and one where $r_2$ is roughly twice the wavelength (around 768 nm). Similarly, the curve for 520 nm has a peak where $r_2$ is near the wavelength (520 nm). If you were to extend the plots to higher wavelength, you would presumably see a second peak

somewhere around 1040 nm.

*Exactly. I updated our statement to adopt some of your language to more clearly communicate this phenomenon. Thank you for the suggestion.*

Droplets with different sizes will have slightly different H2O vapor pressures because it will be impacted by the degree of curvature of the surface, but I would expect the composition of two droplets under the same conditions (temperature and ambient H2O concentration) to have relatively comparable compositions.

*Our original statement was unclear so thank you for raising this. The intent of this question was "what if 1 mode is sulfuric acid and the other is smoke?" The text was updated to remove this ambiguity.*

We can also note from Figure 17 that the values of re from this study are systematically higher than the v7.0 results, which is perhaps another indication that this approach yields a high bias, particularly at smaller mode radius, as was (to me) suggested by the sensitivity study results in Figure 7.

*Agreed, there is a systematic offset between the 2 products. However, the 2 are within the specified uncertainties of the v7.0 products (as discussed within the paper). When doing these types of comparisons the question always becomes: which one is correct? The best we can say is that the 2 are within the reported uncertainty.*

It is surprising that you would choose not to mention Australian Black Summer (or Australian New Year, as it is also called) wildfires from 2019/2020...

*Dr. Boone is correct in that this was a missed opportunity. We have added a brief discussion regarding the Australian fire as well as the recommended warning to the reader.*

No comparison is made between the SAGE II results (Figures 15 and 16) and the SAGE III/ISS results (Figures 18 and 19)...

*The requested comparison was performed and a single figure (for the southern hemisphere) was added. Additional discussion was added to the paper as well. Briefly, we performed the requested hybrid intercomparison wherein we calculated PSD parameters using the SAGE III/ISS data with the 5/6/15 conditions as well as condition 0. Differences between these 2 products were compared. In general, the agreement between the 2 versions is within $\approx 30\%$ for all products within the main aerosol layer. However, there are larger differences (especially for number density) as discussed in the text.*

The difference between the peak altitude near the equator versus that at higher latitudes is not a direct indication of aerosol sedimentation. As was mentioned earlier in the paragraph, Brewer Dobson circulation will naturally transport air to lower altitude as it moves poleward, even in the absence of sedimentation. To get a sense of the descent, you should observe the location of the > 300 nm plume at a particular latitude (e.g., 30 S or 45 S) and note how it changes over time.

*Good point. The context of this paragraph points the reader to Brewer-Dobson circulation,*

so the mention of sedimentation was a non-sequitur on my part. This statement was revised to remove ambiguity.

It depends on how you define the extent of the atmosphere, I suppose. For measurement purposes, I would define the overall height of the atmosphere according to the range over which atmospheric extinction contributes significantly to the signal. With that consideration, I would say you are looking through hundreds of kilometers, not thousands.
Statement revised.

This term was never explicitly defined. Looking at Equation 2, some people might interpret distribution width as $\sigma$, while others might interpret it as $\ln(\sigma)$. The latter choice might be the more logical interpretation. $\sigma$ would be literally interpreted as the width of the distribution in $\ln(r)$ space. $\sigma$ is defined in the paper as the geometric standard deviation, and the text shifts into referring to it as distribution width without comment.
The text was updated to remove ambiguity.

Ideally, when determining a statistical projection of the solution set, your 'basis set' would span the full set of expected conditions. At low temperatures and/or high H2O concentrations, wt% H2SO4 can be 50% or lower (Bernath et al 2023, https://doi.org/10.1016/j.jqsrt.2023.108520). These would fall outside the chosen basis set.
Agreed. However, as discussed above, inclusion of a sulfuric acid estimate would go a long way in mitigating these assumptions. The SAGE III/ISS water vapor product is currently being released and the temperature product is still in development. However, incorporating this capability would be a significant upgrade to the current algorithm.

Traditionally, it would be written as "one can infer" Change made.

Typo. "...how well the two matched"? Corrected

LUTs Corrected

situations -> situation Corrected

$\sigma 2=1.05$ Corrected

k520/k1020 $\leq$ 1.4 Corrected

And Figures 15 and 16 Corrected

influence -> influenced Changed

deposition -> sedimentation? Changed

were -> was (plume is singular). Corrected

2 -> two Corrected

---

## Author Response (AR2)

We would like to again thank Dr. Boone for his close reading of the manuscript and valuable feedback. Our responses are provided below (blue) to the reviewer's comments (black).

Line 330: time -> times
Corrected

Line 363: "product of resonances in the Mie scattering"
I felt it could be specified that the resonances visible in the figure were for r_2 = lambda and 2*lambda
The requested detail was added to the text.

Looking at r_m in Table 9, the percent difference in P_50 tells us that the median value determined for mode radius is larger when using cond0 (by about 28% in the NH and 15% in the SH). Using cond0 appears to dramatically overestimate r_m for small particles (P_10, the radius for which 10% of the measured values are smaller, is about a factor of three larger for cond0). Does the value of P_90 (the radius for which 90% of the measurements were smaller) indicate a low bias for the largest aerosol radii when using cond0 (the value of P_90 for r_m is 21% higher in the NH and 24% higher in the SH when using cond0)? The text suggests there is "good" agreement for larger particles, so that equates to agreement within 25% is deemed to be good agreement? I'm not saying it isn't, just pondering the implications.

This comment, as I read it, takes into question what qualifies as "good" agreement. Historically, aerosol comparisons that fall within ±25% have been categorized as "good" agreement. Personally, I find this level of agreement to be wanting, but I also recognize the difficulties in doing aerosol comparison. That said, these "historical" comparisons were done with differing instruments and differing sampling volumes whereas here we have the *exact same data* so it is not unreasonable to expect very similar results. However, one must realize the limited information content we are working with when using cond0, which results in a under-constrained problem, which leads the differences shown in Table 9. I would argue that further evaluation of the cond0 performance may be warranted in order to better define the applicability of this algorithm to SAGE II data under non-elevated conditions. What this table demonstrates is the same conclusion drawn by Thomason et al. 2008: this type of algorithm tends to work well under elevated aerosol load but struggles under background conditions. Finally, Dr. Boone's interpretation of the table is correct, that there is an apparent low bias in cond0 when particles are large.